# An amphioxus neurula stage cell atlas supports a complex scenario for the emergence of vertebrate head mesoderm

Xavier Grau-Bové [1,9], Lucie Subirana [2,9], Lydvina Meister[2], Anaël Soubigou [2], Ana Neto[3], Anamaria Elek[1,4], Silvia Naranjo [3], Oscar Fornas [5,6], Jose Luis Gomez-Skarmeta[3], Juan J. Tena [3], Manuel Irimia[1,4,7], Stéphanie Bertrand [2,8] ✉, Arnau Sebé-Pedrós [1,4,7] ✉ & Hector Escriva [2] ✉

The emergence of new structures can often be linked to the evolution of novel cell types that follows the rewiring of developmental gene regulatory sub-networks. Vertebrates are characterized by a complex body plan compared to the other chordate clades and the question remains of whether and how the emergence of vertebrate morphological innovations can be related to the appearance of new embryonic cell populations. We previously proposed, by studying mesoderm development in the cephalochordate amphioxus, a scenario for the evolution of the vertebrate head mesoderm. To further test this scenario at the cell population level, we used scRNA-seq to construct a cell atlas of the amphioxus neurula, stage at which the main mesodermal compartments are specified. Our data allowed us to validate the presence of a prechordal-plate like territory in amphioxus. Additionally, the transcriptomic profile of somite cell populations supports the homology between specific territories of amphioxus somites and vertebrate cranial/pharyngeal and lateral plate mesoderm. Finally, our work provides evidence that the appearance of the specific mesodermal structures of the vertebrate head was associated to both segregation of pre-existing cell populations, and co-option of new genes for the control of myogenesis.

Chordates are an animal clade characterized by the presence of a notochord (in at least one stage of their life cycle)[1] and that include vertebrates, tunicates (or urochordates), and cephalochordates (*i.e.* amphioxus). Even if tunicates are phylogenetically more closely related to vertebrates[2] and share with them some morphological features absent in amphioxus[3], they show developmental modalities and a genomic content and organization that have diverged considerably from the chordate ancestral state[4]. On the other hand, amphioxus exhibit relatively conserved morphological, developmental, and genomic characteristics, and represent a model of choice for studying chordate evolution and the emergence of vertebrate novelties[5,6].

The gastrula of cephalochordates has two germ layers: the ectoderm, which forms the epidermis and the central nervous system, and the internal mesendoderm, which develops into mesodermal

[1]Centre for Genomic Regulation (CRG), Barcelona Institute of Science and Technology (BIST), Barcelona, Spain. [2]Sorbonne Université, CNRS, Biologie Intégrative des Organismes Marins, BIOM, F-66650 Banyuls-sur-Mer, France. [3]Centro Andaluz de Biología del Desarrollo (CABD), CSIC-Universidad Pablo de Olavide-Junta de Andalucía, Sevilla, Spain. [4]Universitat Pompeu Fabra (UPF), Barcelona, Spain. [5]Flow Cytometry Unit, Centre for Genomic Regulation (CRG), The Barcelona Institute for Science and Technology (BIST), Barcelona, Spain. [6]Departament de Ciències Experimentals i de la Salut, Universitat Pompeu Fabra (UPF), Barcelona, Spain. [7]ICREA, Barcelona, Spain. [8]Institut universitaire de France (IUF), Paris, France. [9]These authors contributed equally: Xavier Grau-Bové, Lucie Subirana. ✉e-mail: stephanie.bertrand@obs-banyuls.fr; arnau.sebe@crg.eu; hescriva@obs-banyuls.fr

structures in the dorsal part, and into endodermal structures in the ventral region[7]. Unlike vertebrates, the mesoderm is first simply divided during neurulation into the axial territory forming the notochord, and the paraxial domain that becomes completely segmented into somites from the most anterior to the posterior part of the embryo. In vertebrates, in addition to the notochord and somites, the mesoderm is subdivided into other territories: the lateral plate mesoderm in the trunk that forms several structures among which part of the heart and circulatory system, blood cells, fin buds or excretory organs[8]; and the prechordal plate (axial) and cranial/pharyngeal (non-axial, unsegmented) mesoderm in the anterior region that form head muscles and part of the heart[9,10]. If we consider that the amphioxus mesoderm organization could resemble that of the chordate ancestor, these mesodermal territories represent vertebrate-specific traits that contributed to the acquisition of particular structures, including the complex vertebrate head.

Although the entire paraxial mesoderm of amphioxus is segmented into somites, which first form epithelial spheres during neurulation, these segments elongate during development towards the ventral region where the cells intermingle between the endoderm and the epidermis[11]. The orthologs of several genes expressed in the vertebrate lateral plate mesoderm (or derivatives) are expressed in this ventral region of the amphioxus somites, suggesting homology between these two embryonic territories[11–14]. This hypothesis was further supported by transgenesis experiments showing that the regulatory region of *draculin*, a gene specifically expressed in the entire lateral plate mesoderm of zebrafish, is able to drive expression of a reporter gene in cells of the ventral region of amphioxus somites[15]. On the other hand, two studies, based on electron microscopy and in situ hybridization, suggested the compartmentalization of amphioxus somites not into a myotome part and non-myotome part, but into four regions that would be homologous to the myotome, the sclerotome, the dermomyotome and the lateral plate mesoderm of vertebrates[16,17]. Finally, gene expression and functional data also showed that the first amphioxus somite pair differs from the other somites and that it contributes to the formation of the excretory organ of the larva and of putative hematopoietic cells[14,18,19].

Concerning the axial mesoderm, although the notochord extends along the entire anteroposterior axis of the amphioxus, we and others have shown that the anterior region exhibits cellular behaviors and gene expression that differ from those of the central and posterior notochord[20–22]. Based on these observations and our previous work on the control of somitogenesis in amphioxus, we have proposed a multi-step scenario for the evolution of the vertebrate anterior mesoderm[22,23]. The first step consists of the segregation of the ventral mesoderm from the paraxial mesoderm and the loss of its segmentation. This implies that the ventral part of amphioxus somites is homologous to the vertebrate lateral plate mesoderm as previously suggested[11–17]. The second step corresponds to the loss of the paraxial mesoderm in the anterior part of the embryo. This would have enabled the relaxation of the developmental constraints imposed by the anterior somites and the remodeling of the axial and lateral plate mesoderm, resulting in the appearance of the prechordal plate and cranial/pharyngeal mesoderm. This would mean that i) the cranial/pharyngeal mesoderm has a lateral rather than paraxial origin, and partly shares a common developmental program with the amphioxus anterior somites and ventral part of posterior somites, and ii) the prechordal plate is in part homologous to the amphioxus anterior notochord.

Here we sought to explore the evolutionary origin of the vertebrate head mesoderm from a cell type perspective. In order to compare embryonic cell types between amphioxus and vertebrates, we conducted a scRNA-seq analysis of the *Branchiostoma lanceolatum* neurula (N3)[24,25]. The neurula stage shows the highest global transcriptional similarity with vertebrates[26], corresponding to the chordate phylotypic stage[27], and our cell atlas uncovers the gene expression signatures of most of the previously described embryonic territories at this stage. Concerning the mesodermal compartments, we found a cell population localized in the anterior part of the neurula and with a mixed profile between endoderm and notochord, supporting the existence of a transient prechordal plate-like structure in amphioxus[22,28]. We also show that cells of the first somite pair form a population with a transcriptomic profile different from the posterior somites, highlighting the peculiarity of this somitic pair. Moreover, these cells express orthologues of vertebrate genes expressed in both head and lateral plate mesoderm and their derivatives, bringing further support to our evolutionary scenario, and suggesting how, from pre-existing cell populations, new embryonic territories might have emerged in vertebrate anterior mesoderm. Finally, transgenic zebrafish experiments using regulatory regions of amphioxus *Gata1/2/3*, *Tbx1/10* and *Pitx* also supports the lateral origin of the cranial/pharyngeal mesoderm and gives insights into how genes that were presumably not controlling muscle formation in the chordate ancestor were co-opted as master genes of the myogenesis program in the vertebrate head.

## Results and discussion
### A cell atlas of the amphioxus neurula stage embryo

To build a transcriptional cell atlas of the amphioxus neurula stage (N3), we applied MARS-seq[29] to embryos at 21 hpf (hours post-fertilization, at 19 °C) (Fig. 1a). Briefly, cells were dissociated and alive single cells (calcein positive, propidium-iodide negative) were sorted into 384-well plates, followed by scRNA-seq library preparation. At this developmental stage, the embryo is made of around 3,000 cells and we sampled in total 14,586 single-cell transcriptomes, representing approximately a five-fold coverage. These cells were grouped into 176 transcriptionally coherent clusters (referred to as "metacells")[30] (Fig. 1b, Supplementary Fig. 1a). Metacells were further assigned to a tissue/cell type by using transcriptional signatures of known marker genes: epidermis, endoderm, mesoderm, muscular somite and neural (Fig. 1c, Supplementary Fig. 2a). The proportion of cells assigned to each structure/germ layer was overall consistent with cell counting in 3D embryos reconstructed using confocal imaging of labeled nuclei followed by image segmentation, with more than half of the cells belonging to the epidermis (Fig. 1d).

Gene expression signatures across epidermal metacells shows that this tissue is not homogenous. For example, we recognized anterior epidermal cells (i.e. expressing *Arpd2, Fgfrl, Fzd5/8, Pax4/6*)[20,31–33], posterior cells (*Cdx, Tbx6/16, Wnt3*)[34–36] and subpopulations of potential epidermal sensory cells (*Delta, Elav, Tlx*)[37–39]. Among the neural metacells, we identified several metacells corresponding to the presumptive cerebral vesicle (anterior central nervous system, *Otx*[40]). Concerning the mesodermal cell populations, we could assign several metacells to the notochord (*Cola, Foxaa, Mnx, Netrin*)[41–44] and tailbud compartments (*Nanos, Piwil1, Vasa, Wnt1*)[36,45,46]. In the endoderm, one metacell could be assigned to the ventral endodermal region that later develops into the endostyle and the club-shaped gland (*Foxe, Nkx2.5*)[12,47]. We further validated our atlas by analyzing by in situ hybridization the expression of several genes with undescribed patterns at this stage, including genes enriched in neural plate (*Tcf15-like*), endoderm (*PLAC8 motif-containing protein 1*), anterior epidermal (*Tmprss15*), presumptive cerebral vesicle (*Calcitonin Family Peptide 1* (*Ctfp1*)), notochord (*Tenascin*) or tailbud (*Notum*) populations (Fig. 1e and Supplementary Fig. 2a, b). We also verified the cell type identities of the metacells by identifying homologous cell populations in the neurula stage of *B. floridae*[48], another amphioxus species, based on the pattern of shared transcription factors (TFs) (Supplementary Fig. 2c). Overall, our single-cell transcriptomic atlas uncovers the diversity of cell states associated to each major germ layer in the amphioxus neurula. This atlas can be browsed interactively in a dedicated web app: https://sebelab.crg.eu/amphioxus-neurula-21hpf/.

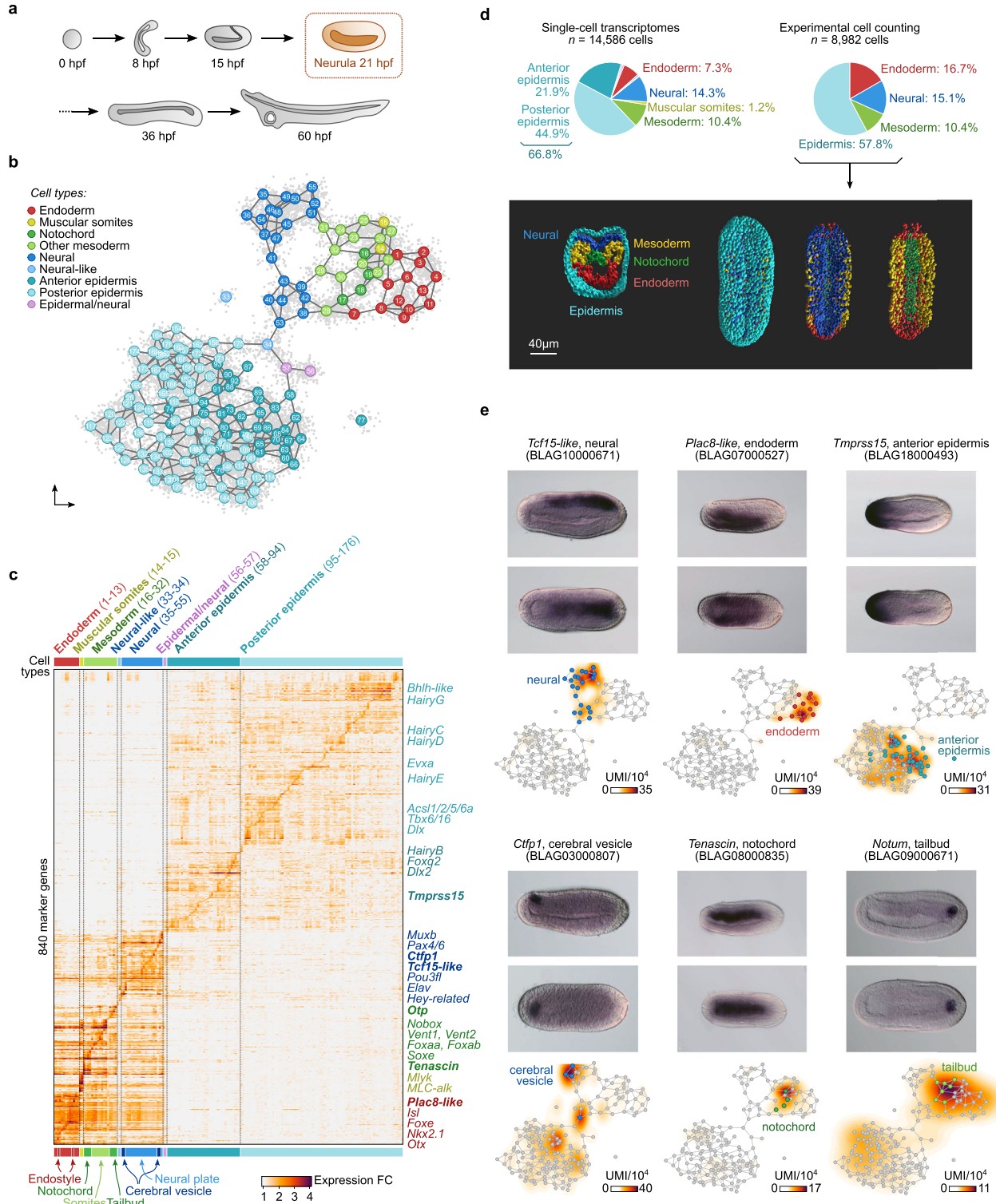

**Cross-species comparison of neurula stage embryonic tissues**

To gain insights into the evolutionary affinities of amphioxus neurula stage tissues, we compared aggregated expression profiles of the different structures and tissues with those of other chordates, using published developmental single-cell atlases for *B. floridae*[48], *Ciona intestinalis*[49], *Xenopus tropicalis*[50], *Danio rerio*[51] (Fig. 2a, b) and *Mus musculus*[52] (Supplementary Fig. 3). We focused our comparative analysis on stages approximately corresponding to the amphioxus neurula stage[26] and used single-cell expression profiles similarly grouped into embryonic tissues.

In general, we find strong similarities between the two amphioxus species (which diverged *ca.* 100 Mya[53]), with reciprocal matches for most cell types across species (Fig. 2a). As expected, the similarity between our *B. lanceolatum* neurula and the various cell types along the *B. floridae* time course is highest in the neurula stages of the latter (Fig. 2b). These results are in line with the validation of our cell type

**Fig. 1 | Amphioxus neurula cell type atlas. a** drawings of Mediterranean amphioxus developmental stages from the egg to the larva (with one open gill slit) stage. The developmental time (hours post fertilization, hpf) is given for embryos raised at 19 °C, and we highlight the neurula stage presented in this study (21 hpf). **b** Two-dimensional projection of cell clusters (metacells) using a force-directed layout based on the co-clustering graphs for individual cells (see *Methods*). Metacells are color-coded by cell type. **c** Normalized fold change expression of top variable genes (rows) per metacell (columns, grouped by cell type). For each metacell, we selected up to 30 markers with a minimum fold change ≥ 2. Selected gene names from known markers, used to annotate each cell type, are indicated to the right of the heatmap. Genes in bold case are shown in panel (**d**). **d** Pie charts depicting the fraction of cells mapped to each cell type among the cell transcriptomes and the cell counting experiment (top); and the 3D reconstruction with assignment of nuclei to each germ layer (bottom). A transverse section is shown on the left, and dorsal views with anterior to the top on the right (full, without epidermis nuclei, without epidermis and neural cells nuclei. **e** Expression profile of previously unknown marker genes for specific cell types (neural, endoderm, anterior epidermis, presumptive cerebral vesicle, notochord, and tailbud) analyzed by in situ hybridization (ISH, top, with anterior to the left and dorsal to the top in side views, $n = 10$ embryos) and corresponding two-dimensional expression maps (bottom, based on the same layout as panel **b**). Gene expression is shown as density maps representing UMI counts (per 10,000 UMIs) in each cell.

classification based on the identification of cell populations with matching TF usage across both species (Supplementary Fig. 2c).

In the pairwise comparisons with *C. intestinalis*, *X. tropicalis* and *D. rerio*, the notochord showed the strongest transcriptional similarity and shared expression of the TFs *Brachyury2* (*T*), *Foxaa* and *Foxab* (*Foxa1* and *Foxa2*) (Fig. 2c). Conversely, in spite of sharing the same core TFs, the mouse and the amphioxus notochords exhibited low global transcriptomic similarity (Supplementary Fig. 3). This lack of conservation compared to mouse probably reflects the different structural roles that the notochord can play during development in amniote and non-amniote vertebrates. Likewise, amphioxus differentiated muscular somites resemble muscle/skeletal muscle in tunicates and the three vertebrates (Fig. 2a), albeit with different sets of TFs between species (Fig. 2c). In contrast, the non-muscular part of amphioxus somites resembles vertebrate presomitic mesoderm and shares expression of the TFs *Foxc* (*Foxc2*), *Snail* (*Snai2*) and *Hox3* (*Hoxa3*) (Fig. 2c).

Amphioxus neural cells also resemble vertebrate neural populations and co-express neural TFs like *Soxb1c* (*Sox2*), *Soxc (Sox4)* and *Neurogenin* (*Neurog3*). These same TFs are also shared by tunicate neural cells, but the overall transcriptome does not show similarity with amphioxus neurons. The opposite is true for the endodermal transcriptome: amphioxus endoderm transcriptome matches that of tunicates, but not vertebrate endoderm, although the TFs *Foxaa* and *Foxab* (*Foxa1* and *Foxa2)* are expressed in all of them (Fig. 2c). Finally, the amphioxus anterior epidermis looks more distinct than the posterior one. Among vertebrate epidermal cells, its most similar pairs are secretory cells both in *Danio* and *Xenopus* (termed "Goblet cells" there). But it also hits different mesodermal tissues in *Danio* (e.g. the endothelium). On the other hand, the amphioxus posterior epidermis is broadly similar to many epidermal cell types of the two vertebrates, most notably the epidermal progenitors and ionocytes.

When examining the lists of shared markers between transcriptionally similar embryonic tissues/cell types (Supplementary Data 1), we observed a general overrepresentation of transcription factors (TFs) and chromatin factors compared with effector genes, as expected when comparing undifferentiated cell populations.

## The accessible chromatin landscape of amphioxus neurula stage

To interrogate the regulatory logic underlying the observed cell-specific transcriptomes, we performed bulk ATAC-seq experiments in neurula-stage embryos. We defined a total of 51,028 ATAC-seq peaks and assigned them by proximity to 19,069 genes (median 2,05 peaks per expressed gene) (Supplementary Fig. 1b–h). We then grouped these peaks according to the expression pattern of the associated genes and conducted motif enrichment analysis on these regulatory element groups, using a combination of de novo inferred and known motifs (see Methods). This analysis revealed 317 distinct motifs with significant enrichments in specific cell populations (Fig. 3a).

The identified motifs are consistent with known TF regulators in amphioxus and other metazoans and, in addition, motif enrichments often parallel the expression of the associated TFs (Fig. 3b,

Supplementary Fig. 4). For example, in peaks assigned to epidermal genes, we found enrichment for motifs like Dlx, Grhl, Klf1/2/4, Rfx1/2/3 or Tfap2, coincident with the expression of *Dlx*, *Klf1/2/4* and *Tfap2* in epidermal metacells (Fig. 3b). Interestingly, these TFs are part of the in silico reconstructed gene regulatory network controlling epidermis development described in amphioxus[54] and are known epidermal fate determinants in vertebrates[55–58] (Fig. 2c).

In endodermal cells, we found a slight but non-significant enrichment of a Fox motif in the regulatory regions of endodermal marker genes, possibly linked to the expression of *Foxaa* and *Foxab* in these tissues (Supplementary Fig. 4). Furthermore, in the endoderm and the endostyle, we also observed the coincident expression/motif enrichment of *Gsc* and *Nkx2-5/6*, respectively (Fig. 3b).

Neural cell types exhibited expression of various Sox and Pou family TFs and concomitant enrichment of their associated motifs, including *Soxc* (*Sox4*), *Soxb2* (*Sox14*), *Soxb1c* (*Sox2*, although its motif enrichment is only significant in the anterior archencephalon at $p < 0.01$), and *Pou3fl* (*Pou3f4*, with significant motif enrichment in the Di-Mesencephalic primordium[20] and neural tailbud cells; Fig. 3b). The neural specificities of these TFs appear to be conserved across vertebrates (Fig. 2c) and SoxB1 and Pou3f family factors have been proposed as potential major regulators of nervous system development in amphioxus[54]. The activity of *Pou3fl* (*Pou3f4*) in neural tailbud cells is also consistent with the function of TFs from these families in the maintenance of stemness in vertebrates[59]. This is also the case for the *Myc/Max* HLHs in the non-neural tailbud population, as observed in mouse[60].

The peaks associated to genes overexpressed in non-muscular somite cell populations are enriched in T-box motifs, consistently with the expression in our dataset of various TFs of this family such as *Eomes/Tbr1/Tbx21*, *Tbx15/18/22*, and *Brachyury2* (Fig. 3b) and with previously reported expression of these genes in forming somites[61–63]. The muscular somite population peaks are enriched in motifs shared with the non-muscular somite, but are also enriched in motifs for Myogenic Regulatory Factors (MRFs) such as *Mrf4* (*Myf6*), which is also highly expressed in this cell type (Fig. 3b), in line with both the expression of the various amphioxus MRFs described by in situ hybridization[64], and the role of their orthologues in vertebrate myogenesis[65]. The strongest TF-motif association concerns the previously reported notochordal marker *Foxaa* (*Foxa2*) (Fig. 3b)[44], which is also shared with tunicates and vertebrates in our cross-species cell type comparisons (Fig. 2c). Overall, the accessible chromatin landscape of the neurula stage revealed the regulatory motif lexicons underlying amphioxus embryonic cell identities.

## Characterization of neural, endodermal and somitic cell populations

We then focused on the detailed analysis of specific cell populations. To this end, we performed separate clustering of single cells classified as belonging to the endoderm and to the somites (muscular and non-muscular). We also performed such an analysis on single cells of the neural tissue (*sensu stricto*, derived from the neural plate) that is detailed in Supplementary Note and Supplementary Fig. 5.

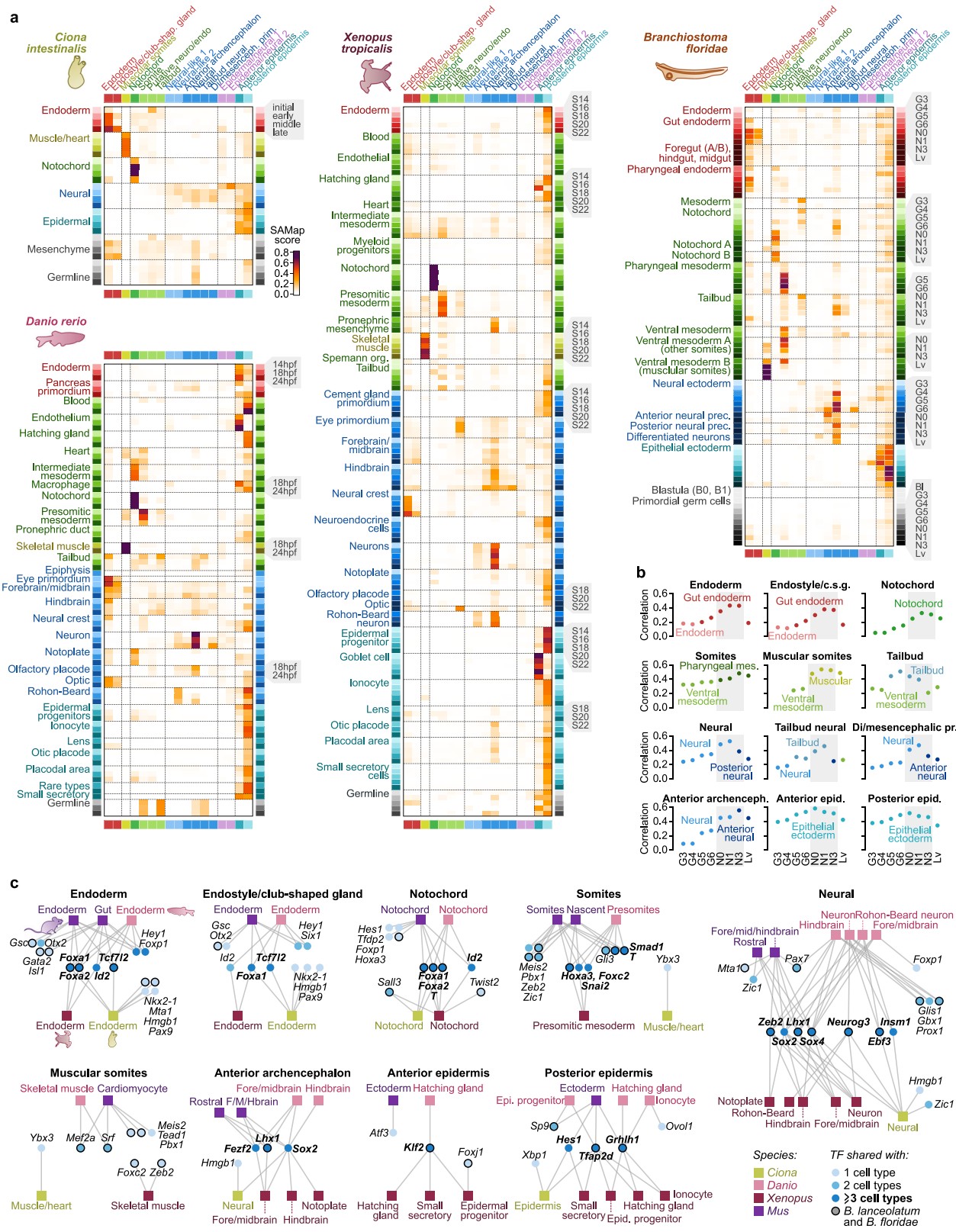

Concerning the endodermal compartment, we could recognize metacells corresponding to the main known territories (Fig. 4a, b and Supplementary Figs. 2 and 6). The expression of the ventral marker *Nkx2.1*[66], together with anteriorly expressed genes such as *Dmbx*, *Fgfrl*, *Fzd5/8* and *Sfrp1/2/5*[20,31,33,67–69] (Supplementary Fig. 6) indicates that metacell 2 corresponds to the ventral anterior endoderm territory whereas metacells 3 and 7 show a combination of marker genes that

are typical of the ventral endoderm that later develops into the club-shaped gland and the endostyle such as *Foxe*, *Nkx2.5*, *Tbx1/10* and *Pax1/9*[12,47,70,71] (Fig. 4a, b and Supplementary Fig. 6). The expression of *Pitx* in metacell 3 suggests that metacells 3 and 7 correspond to the left and right part of this territory, respectively[72].

Posterior to that, metacells 16 and 17 that are characterized by low or no expression of *Soxf* correspond to the first pharyngeal slit anlagen[73]

**Fig. 2 | Cross-species comparison with other chordate developmental datasets.** **a** Comparison between cell type transcriptomes of the amphioxus neurula stage (rows) and matched developmental time-points (rows) in the chordates *Ciona intestinalis* (initial to late tailbud stage), *Danio rerio* (14 hpf to 24 hpf), *Xenopus tropicalis* (S14 to S22 stages), and *Branchiostoma floridae* (including blastula [Bl], four gastrulation stages [G], three neurula [N] stages, and larva [Lv]). Cell type similarity was measured using SAMap scores based on all available pairwise markers (see *Methods*). Cell types are color-coded by cell type or developmental layer (endoderm, mesoderm/muscle, neuroectoderm, ectoderm, and other), and, in the case of the multi-stage chordate datasets, by developmental time-point (color intensity, legend to the side of each plot). **b** Most similar pairs of cell populations between the *B. lanceolatum* neurula (subpanels) and the *B. floridae* developmental stages (color-coded dots in each subpanel, from gastrula [G3] to larva [Lv]).

Similarity is measured using Pearson's correlation coefficient of cell type-level normalized fold change values, using genes with FC > 1.05 in at least one cell type. Only the top *B. floridae* cell type is displayed. Gray boxes highlight the neurula stages, where similarity typically peaks. **c** Graph representation of transcription factors (TFs, circular nodes) shared (i.e. connected by an edge) between amphioxus and homologous cell types in *Ciona*, *Danio* and *Xenopus* (square nodes). Specific TFs are considered to be shared between two cell types if they are significantly overexpressed in both. TFs in bold are shared between all species considered. For amphioxus, we required fold-change > 1.25, and BH-adjusted *p*-value < 0.05. For the matched cell types from other species, we required significant overexpression in at least one of the developmental time-points considered. A complete list of genes shared between all pairs of cell types is available in Supplementary Data 1. Di/mesenceph. prim= Di/mesencephalon primordium.

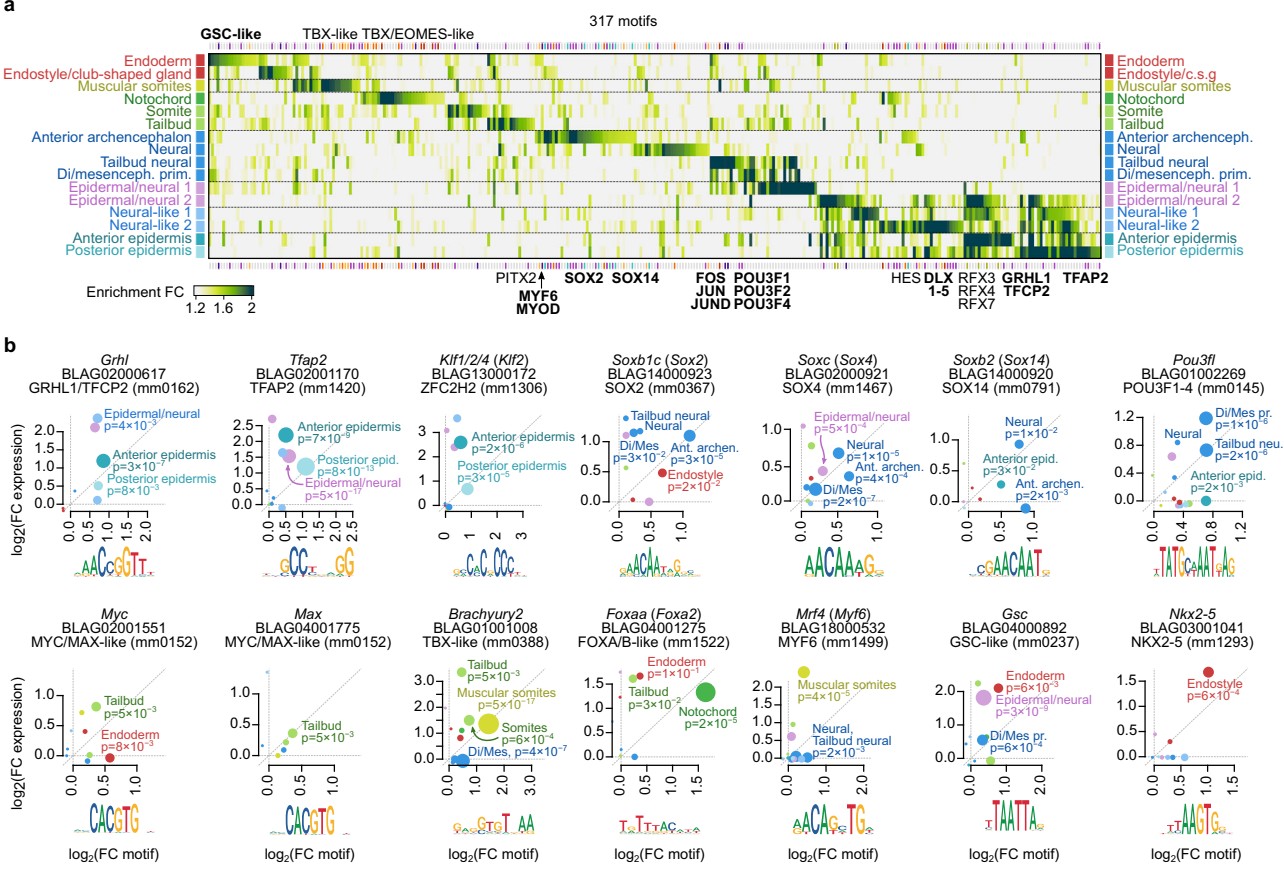

**Fig. 3 | Regulatory landscape of neurula cell types. a** Heatmap representing the enrichment of specific TF binding motifs (columns) in the regulatory regions of genes in each cell type of the amphioxus neurula (rows). Names from selected motifs are indicated next to the heatmap (in bold those that also appear in panel b). The amphioxus motif library was obtained by merging experimentally determined vertebrate motifs from CIS-BP with de novo inferred motifs for each amphioxus cell type, and removing redundancy (see Methods). Therefore, motif names do not represent specific amphioxus TFs, but rather sequence similarity with motifs of vertebrate homologs. The regulatory regions associated with each gene were

obtained from a bulk ATAC-seq experiment. **b** Examples of cell type-specific amphioxus TFs whose expression levels (vertical axis, as log2(FC)) match the enrichment of associated motifs (horizontal axis; shown below as information content logos). Circle size is proportional to the BH-adjusted −log10(*p*) binomial test of motif enrichment comparing motif occurrence near cell type-specific and unspecific genes (see Methods; exact *p*-values of these tests are shown for significant enrichment at *p* < 0.01, see Source Data for a complete list). Di/mesenceph. prim= Di/mesencephalon primordium.

while metacell 14 expresses both *Irxc* and *Foxaa*, a combination specifically observed in a region that is just behind it[44,74] (Fig. 4a, b and Supplementary Fig. 6). Metacells 5 and 11 express *Pax1/9* but no ventral markers and could correspond to the dorsal mid endoderm region[70] (Fig. 4a, b). Metacells 8 and 12 show a very similar profile with an enrichment in transcripts of mid/posterior endoderm markers such as *Nkx2.2*, *Foxaa*[44,75] (Supplementary Fig. 6), and *Fabp3/4/5/7/8/9/11/12*, a marker described here for the first time (Fig. 4a, b). Metacell 12

additionally expresses *Gata4/5/6*, indicating that the corresponding cells are more ventral than those from metacell 8[14] (Supplementary Fig. 6). Metacell 9 has a transcriptional profile similar to that of metacells 8 and 12 combining expression of the mid/posterior marker *Foxaa*[44] (Supplementary Fig. 6) and absence of *Pax1/9* expression[70] (Fig. 4a, b). The posterior marker *Wnt8*[76] is expressed in metacells 4 and 6 with metacell 4 also expressing the ventral marker *Gata4/5/6*[14], and, hence, representing the ventral posterior territory (Supplementary Fig. 6).

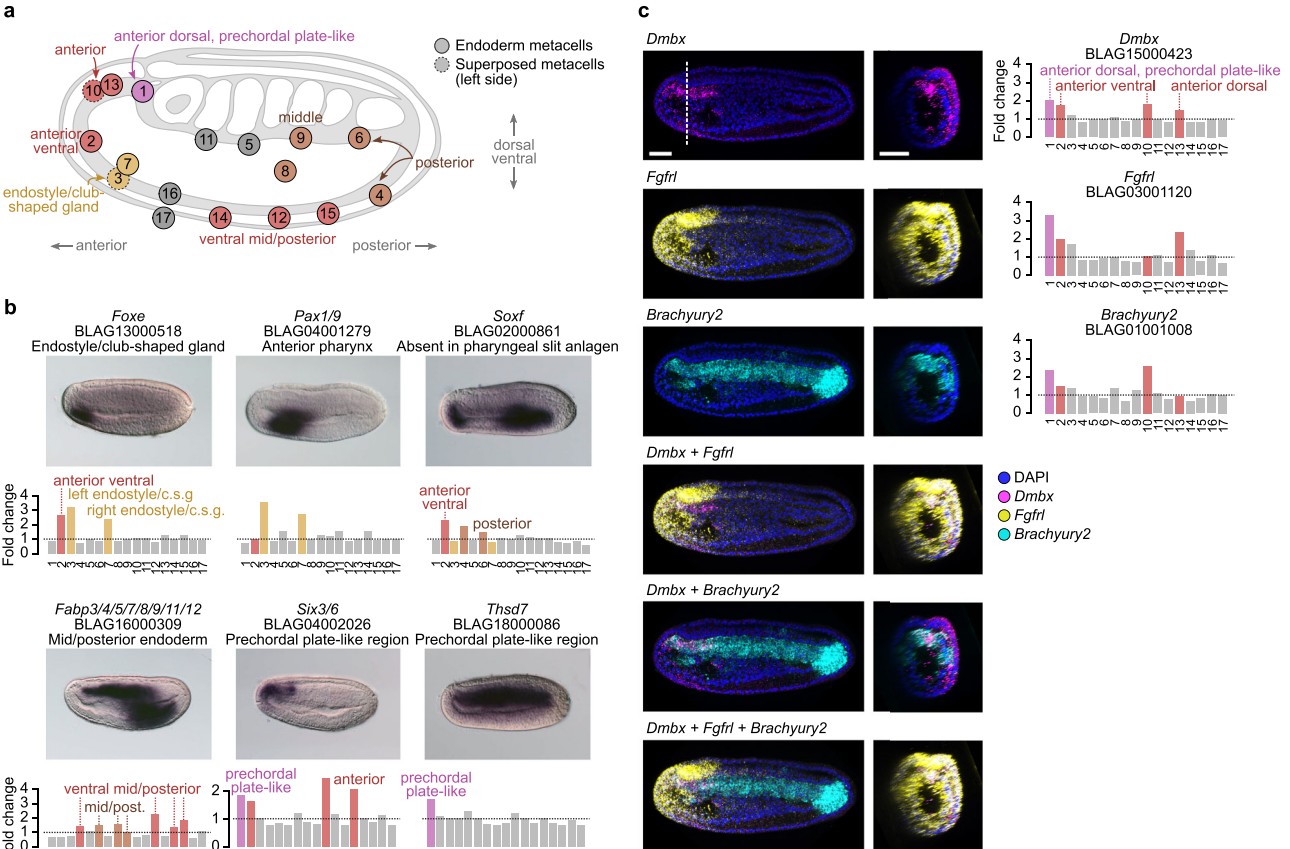

**Fig. 4 | Endoderm metacell subclustering. a** 2D projection of endodermal metacells on a side view scheme of an amphioxus neurula stage embryo with anterior to the left and dorsal to the top. **b** Gene expression as normalized fold changes of selected gene markers and corresponding ISH (n = 10 embryos). Gene expression is shown as density maps representing UMI counts (per 10,000 UMIs) in each cell, with cells positioned in the vicinity of their corresponding metacells. **c** Multiple in situ hybridization for *Dmbx*, *Fgfrl* and *Brachyury2* (n = 5 embryos).

Lateral views are shown on the left, with anterior to the left and dorsal to the top. Optical cross-sections at the level of the dotted line are also shown. The labeling intensity is stronger on one side due to the thickness of the embryo that was imaged on one side after mounting. The three genes are coexpressed in the anteriormost region of the dorsal mesendoderm (metacell 1). Normalized fold change expression of each marker is shown on the right. Scale bar: 25 μm.

Finally, metacells 1, 10 and 13 are characterized by an enrichment in anterior markers *Dmbx*, *Fgfrl*, *Fzd5/8* and *Sfrp1/2/5*[31,33,67–69] as well as *Six3/6*, *Six4/5* and *Zic*[77,78] (Fig. 4a, b and Supplementary Fig. 6). They show a transcriptional signature of the anterior dorsal mesendoderm, a region which is continuous with the notochord per se posteriorly, and which is continuous laterally with the endoderm per se. Metacell 1 is also expressing *Thsd7* described here (Fig. 4a, b), together with *Brachyury2*, *Pax3/7* and *Zeb*[54,63,79] and lacks *Nkx2.1* expression[80] (Supplementary Fig. 6) suggesting it represents the axial part of this region, whereas metacells 10 and 13, expressing *Nkx2.1*, would correspond to the paraxial more ventral portion that latter form the left and right Hatschek's diverticula[80] (Supplementary Fig. 6). Therefore, metacell 1 represents a potential prechordal plate-like territory showing a transcriptomic profile characterized by anterior and axial markers together with endodermal markers.

We validated the existence of this cell population by undertaking multiple in situ hybridization for *Dmbx*, *Fgfrl* and *Brachyury2*, showing a clear overlap of expression in the anterior tip of the dorsal axial mesendoderm (Fig. 4c). Such territory was already proposed to exist in amphioxus based on both cell behavior and gene expression of several marker genes[21,22,28] but our data highlight the strong difference in its transcriptomic profile compared to the other notochord cells, reinforcing the idea that ancestral chordates possessed a prechordal plate-like region that later evolved specific functions in vertebrates.

Re-clustering of cells assigned to the somites resulted in 12 metacells (Fig. 5a, b and Supplementary Figs. 2 and 7). As expected, we found a population (metacell 8) with a profile typical of the muscular part of somites that starts to differentiate, characterized by the expression of *Mef2*, *Lmo4*, several MRFs, together with *MLC-alk*[64,81,82] (Supplementary Fig. 7) and the *Titin-like* marker described here (Supplementary Figs. 2 and 7). Metacells 7 and 9 have similar profiles and also express *Titin-like* and several MRFs[64] (Supplementary Figs. 2 and 7) together with *Brachyury2*, *Delta*[39,63] and the newly described gene *Twist-like* (Supplementary Figs. 2 and 7). They hence correspond to the last somites that have just been formed, with metacell 7 more posterior as indicated by the expression of *Wnt1* or *Wnt4*[76].

More posteriorly, metacell 5 is characterized by the expression of newly described tailbud gene markers such as *Bicc*, and *SF2 family helicase* (Supplementary Figs. 2 and 7), together with *Nanos*, *Otp*, *Vasa*, and *Wnt1*, *4* and *6*[20,46,76] but also expresses *Brachyury2* and *Mrf4*, a combination corresponding to the tailbud somitic part[63,64] (Supplementary Fig. 7). Metacells 4 and 6 also express tailbud markers but do not express MRF genes. Moreover, metacell 4 is characterized by an enrichment in transcripts of the ventral markers *Gata1/2/3* and *Vent1/Vent2*[11,14,83] (Fig. 5a, b and Supplementary Fig. 7).

Concerning the four putative compartments of formed somites proposed by Young and colleagues[17], we could only recognize, apart from the myotome region (clusters 7, 8 and 9), a non-myotome region (corresponding to cell clusters 2 and 11) enriched in markers

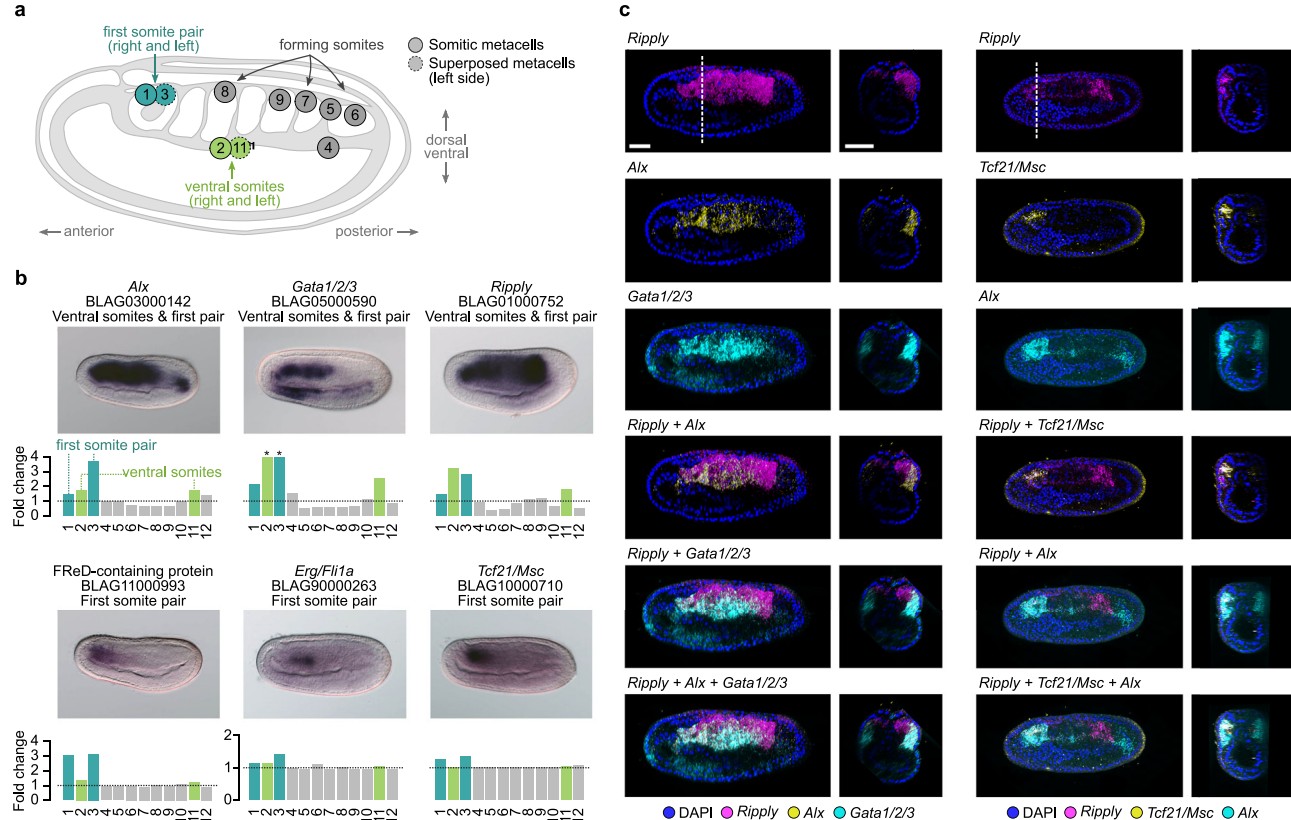

**Fig. 5 | Somite metacell subclustering. a** 2D projection of somitic metacells on a side view scheme of an amphioxus neurula stage embryo, with anterior to the left and dorsal to the top. **b** Gene expression as normalized fold changes of selected gene markers and corresponding ISH (n = 10 embryos). **c** Multiple in situ hybridization for *Ripply+Alx+Gata1/2/3* and *Ripply+Tcf21/Msc+Alx* (n = 5 embryos). Lateral views are shown on the left, with anterior to the left and dorsal to the top. Optical cross-sections at the level of the dotted line are also shown. The labeling intensity is stronger on one side due to the thickness of the embryo. *Ripply*, *Alx* and *Gata1/2/3* are coexpressed in the non-muscular part of formed somites (metacells 2 and 11) whereas *Ripply*, *Tcf21/Msc* and *Alx* are coexpressed in the first somite pair (metacells 1 and 3). Scale bar: 25 μm.

expressed in territories described as lateral and/or ventral such as *Alx*, *Gata1/2/3*, *Ripply* and *Vent1/Vent2*[11,14,42,84] (Fig. 5a–c and Supplementary Fig. 7). We validated the existence of these cell populations using in situ hybridization for *Alx*, *Gata1/2/3* and *Ripply*, which showed co-expression of the three genes in the non-myotome region of somites (Fig. 5c). To simplify, we will refer to the non-myotome portion of somites as the "ventral somite" throughout the remainder of the manuscript.

Finally, the most important novelty concerns the first somite pair, which clearly shows a transcriptomic profile divergent from the other pairs. Metacells 1 and 3 correspond to this first pair of somites, with metacell 1 representing the right somite, and metacell 3 the left one (Fig. 5a, b). Indeed, contrary to metacell 1, cells of the latter express the left side marker *Pitx*[72] as well as *Gremlin*, which is expressed in the first left somite at this stage[85] (Supplementary Fig. 7). Both metacells express the anterior marker *Fgfrl*[31] (Supplementary Fig. 7), and three newly described markers: *Erg/Fli1a*, *Tcf21/Msc* and *FReD containing protein* (Fig. 5a, b and Supplementary Fig. 7). They also express the ventral somite marker genes *Alx*, *Gata1/2/3*, *Ripply* and *Vent1/Vent2*[11,14,42,83,84] (Fig. 5a, b and Supplementary Fig. 7). To note, no Wnt genes are expressed in these metacells, whereas the ventral markers are expressed together with *Wnt16*[76] in metacells 2 and 11 that correspond to the ventral region of the formed somites posterior the the first pair (Supplementary Fig. 7). Interestingly, *Erg/Fli1a* is orthologous to *Fli-1* which is implicated in vertebrate hemangioblast development together with *Vegfr* and *Scl/Tal-1*[86]. It has been shown that the amphioxus orthologs of these genes are also expressed in the first pair of somites[14], reinforcing the proposition of homology between this

first pair and the embryonic hematopoietic/angiogenic field of vertebrates that derives from the lateral plate mesoderm. On the other hand, *Tcf21/Msc* is orthologous to *Tcf21/Capsulin* and *Msc/MyoR* are main regulators of head muscle myogenesis in vertebrates, upstream of MRFs[87–89], suggesting that the first somite pair of amphioxus has a profile that resembles both vertebrate head and lateral plate mesoderm. We further validated the co-expression of *Tcf21/Msc* with *Ripply* and *Alx* in the first somite pair (Fig. 5c).

## The evolution of the chordate anterior mesoderm
The most striking feature of the amphioxus neurula highlighted by our data is the presence of three cell populations with a peculiar transcriptional profile: cells of the first left and right somites (metacells 1 and 3, Fig. 5a, b), and cells that could correspond to a prechordal plate-like structure (metacell 1, Fig. 4a, b). The first somite pair in amphioxus has long been proposed as being distinct from the other pairs, and we previously showed that this somite pair is the only one whose formation is controlled by the FGF signaling pathway[22,23,90]. Our molecular atlas additionally shows that the cells of the first pair of somites transcriptionally resemble vertebrate head and lateral plate mesoderm (metacells 1 and 3, Fig. 5 and Supplementary Fig. 7), while the cells of the ventral part of amphioxus somites posterior to the first pair express orthologues of genes expressed in vertebrates lateral plate mesoderm or derivatives (metacells 2 and 11, Fig. 4 and Supplementary Fig. 6). These data support the homology between vertebrate lateral plate mesoderm and amphioxus ventral part of the somites, as well as the ventral origin of vertebrate cranial/pharyngeal mesoderm.

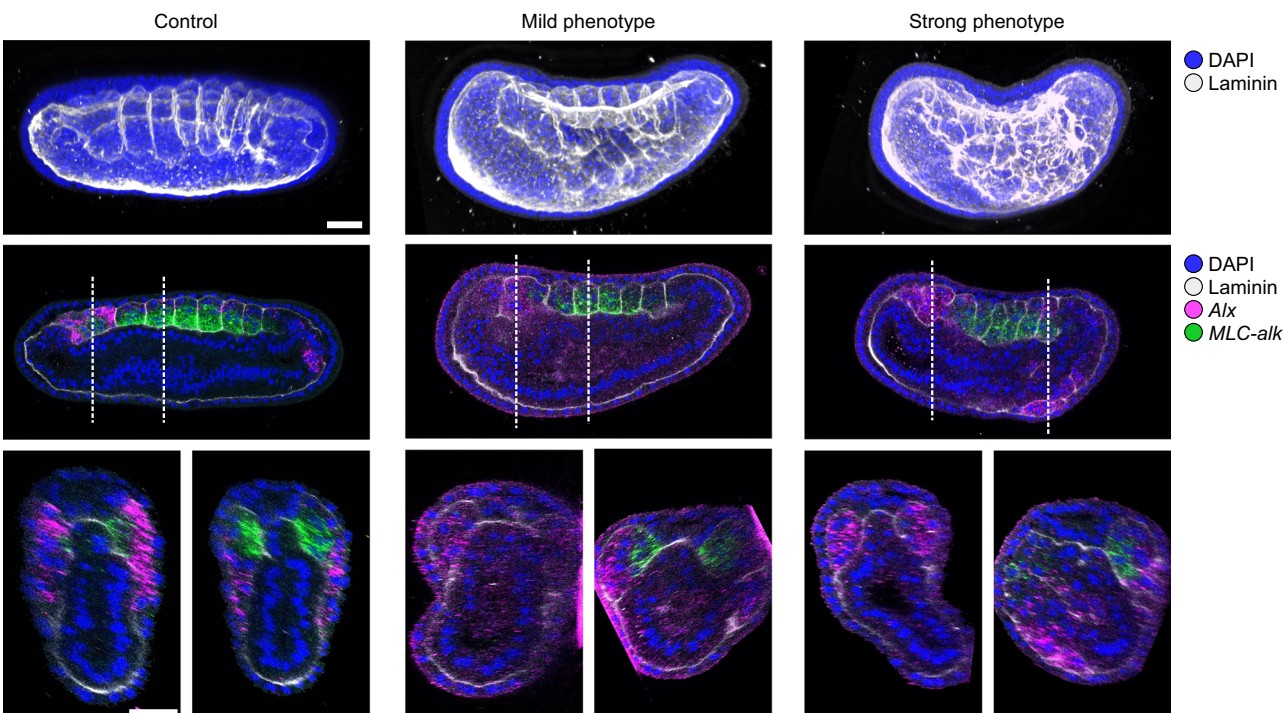

**Fig. 6 | Phenotype of N4 stage embryos after overexpression of a constitutive repressor form of Gata1/2/3.** Control embryos ($n$ = 5), injected embryos with a mild phenotype ($n$ = 5/10) and injected embryos with a strong phenotype ($n$ = 5/10) from two independent injection experiments were processed for double in situ hybridization for *MLC-alk* and *Alx* followed by immunostaining against Laminin. Side views of 3D reconstructions of the nuclei and Laminin labeling are shown on the top, with anterior to the left and dorsal to the top. Sections in side views showing all the chanels are shown below, as well as transverse optic sections at the level indicated by the white lines. *MLC-alk* is expressed in the myotome region of the somites that is little affected. *Alx* is expressed in the ventral region of the somites that is completely disorganized in injected embryos, as revealed by anti-Laminin labeling. Scale bar: 25 μm.

Such proposed homology based on transcriptomic profile should reflect a conserved regulatory logic. Considering homology at the level of gene expression regulation, we reasoned that if our scenario for vertebrate mesoderm evolution supported by our cell atlas is correct, regulatory regions of genes that are active in the ventral region of the somites at the neurula stage in amphioxus could drive expression of a reporter gene in the vertebrate lateral plate and head mesoderm, as a reminiscence of an ancestrally shared regulatory program. Among such genes, *Gata1/2/3* is the transcription factor with the highest enrichment (fold change) in metacells 2 and 11 of the somite reclustering analysis (Fig. 5a, b), which we could assign to the ventral region of the somites. We first decided to test for the function of Gata1/2/3 in amphioxus embryogenesis to validate its role in the development of the ventral region of the amphioxus somites.

We showed that overexpression of a constitutive repressor form (Gata1/2/3-engrailed) induced somite defects clearly visible at the N4 stage (Fig. 6). Indeed, although the dorsal *MLC-alk* expressing region, which later form the muscles, is little affected, the ventral *Alx*-positive part, which normally elongates at this stage, has formed anarchically organized epithelial spheres, surrounded by a basal lamina, which have colonized the ventral region in embryos with the strongest phenotype (Fig. 6). These data demonstrate that Gata1/2/3 plays a major role in the development of this somite compartment and that repression of expression of its target genes induces a loss of coordinated cell-shape changes and migration.

We hence decided to test whether the regulatory elements controlling the expression of amphioxus *Gata1/2/3* at the neurula stage are recognized by any tissue/cell type specific regulatory state in zebrafish, which would point at evolutionary conservation (at least partially) of *Gata1/2/3* regulation. We show that one of the putative regulatory regions selected using ATAC-seq data (Supplementary Fig. 8a) was able to drive the expression of the *eGFP* reporter gene in zebrafish

embryonic regions that partly overlap with the expression of *Prrx1a* in the head mesoderm and finbuds (Supplementary Fig. 8b and Supplementary Fig. 9a). This suggests that *Gata1/2/3* in the chordate ancestor probably already had the potentiality to be recruited during vertebrate evolution for expression in the pharyngeal and lateral plate mesoderm. However, these data still need to be supported by a more precise analysis of reporter expression domains at different developmental stages.

In vertebrates, both the anterior axial (prechordal plate) and pharyngeal/cranial mesoderm structures develop into different muscle populations: the extraocular muscles, and several facial/branchial muscles, respectively[10]. Interestingly, myogenesis in these cells, although it is mediated by the activity of members of the MRF family, is controlled by the upstream factors *Pitx2* (extraocular muscles) and *Tbx1* (pharyngeal muscles) and not by *Pax3/7* and *Six1/2* factors as it is the case for muscles deriving from the somites[10,89,91]. In amphioxus, we previously showed that all the somites form under the control of *Pax3/7*, *Six1/2* and/or *Zic*[23]. Moreover, *Pitx*, the ohnologue of vertebrate *Pitx1*, *Pitx2* and *Pitx3*, has been shown by in situ hybridization to be expressed on the left side of the embryo and in few neurons and is controlling left/right asymmetry[36,72,92], while we observed in our data its expression only in two metacells (3 and 11) in the somite subclustering atlas (Supplementary Fig. 7). On the other hand, *Tbx1/10* has been shown to be expressed long after MRFs in the amphioxus somites[23,71] and we showed in our data a reduced expression in metacell 8 in the somite subclustering atlas (Supplementary Fig. 7), metacell we assigned to the muscular part of the trunk somites, while its expression was not detected in the other metacells expressing MRFs. Furthermore, functional analysis by TALEN mutagenesis or morpholino injection for *Pitx*[93] and *Tbx1/10*[94] respectively, showed that knockout or knockdown of these genes did not alter somite development.

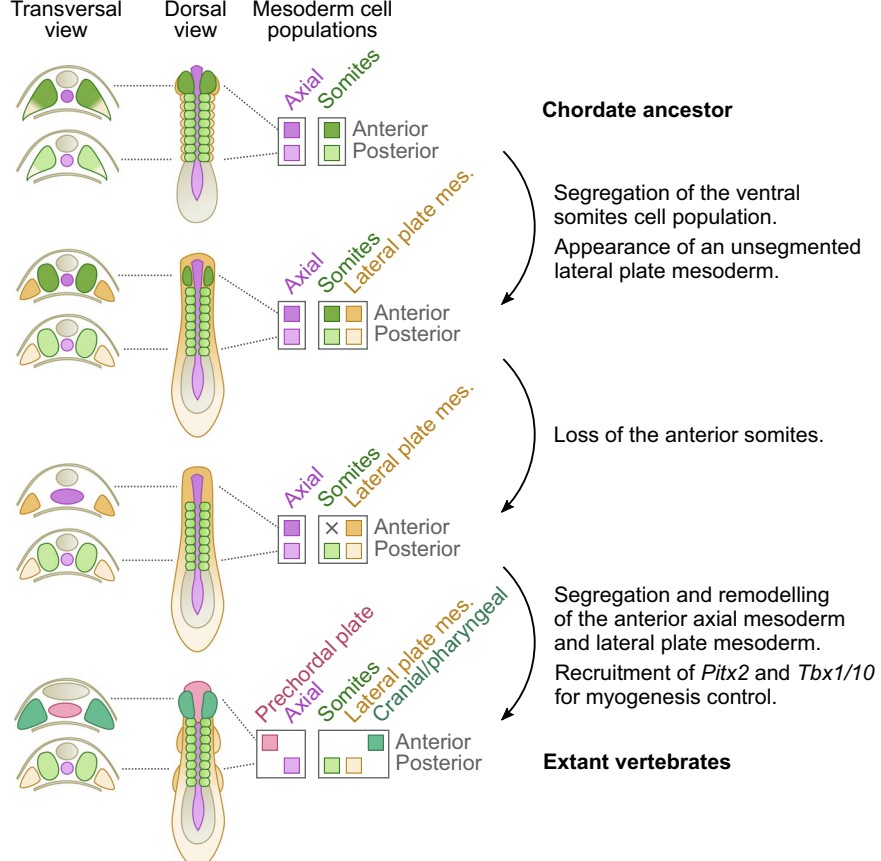

**Fig. 7 | Evolutionary scenario for mesoderm evolution in chordates.** Schemes of putative embryos in dorsal views with anterior to the top are shown, with transverse sections at the level of the anterior and trunk regions on the left. Diagrams on the right represent mesoderm cell populations that were inferred at each step. We propose that the chordate ancestor possessed a mesoderm organized in an axial domain with two cell populations: a prechordal-plate like region in the anterior part (dark purple), and a notochord (light purple) more posteriorly; and a paraxial domain completely segmented into somites (green), containing in the ventral part a cell population homologous to the ventral part of amphioxus somites (orange), and showing heterogeneity between the anterior (dark orange/green) and trunk (light orange/green) regions. During the first step of evolution, we propose that the ventral somite cell populations became independent from the paraxial mesoderm to give rise to the unsegmented lateral plate mesoderm (lateral plate mes., orange). In a second step, the anterior paraxial mesoderm would have been lost (dark green), and we previously proposed that this could be due to a change in the function of the FGF signaling pathway[22,23]. This loss would have led to a relaxation of the developmental constraints imposed by the segmented paraxial mesoderm in the anterior region, enabling remodeling of the tissues of the anterior axial mesoderm (dark purple) and anterior lateral mesoderm (dark orange), which could have evolved into the prechordal plate (pink) and pharyngeal/cranial mesoderm (blue/green). The ability of these new embryonic structures, derived from non-myogenic cell populations, to form muscles, would have been associated with the co-option of *Pitx2* and *Tbx1/10* as master genes of the myogenesis program.

If our scenario of head mesoderm evolution is correct, it implies that *Pitx2* and *Tbx1* were co-opted for the control of myogenesis in the vertebrate head. In order to test this co-option, we investigated the activity of putative regulatory regions of amphioxus *Pitx* and *Tbx1/10* in zebrafish reporter assays. In the case of *Tbx1/10*, one genomic region tested was able to drive the expression of the reporter gene in head and finbud regions that partly overlap with *Prrx1a* expression territories (Supplementary Fig. 8c, d, and Supplementary Fig. 9b). This suggests that *Tbx1/10* in the chordate ancestor probably contained regulatory information that allowed its later recruitment in the vertebrate head mesoderm for a new function as a myogenesis controlling factor. In the case of *Pitx*, one region drove a restricted reporter expression in the zebrafish hatching gland (Supplementary Fig. 8e, f). Interestingly, this structure derives from the anterior prechordal plate and expresses *Pitx2*[95–98], suggesting that the *Pitx* gene in the chordate ancestor already had the potentiality to be recruited in this mesoderm region during vertebrate evolution. However again, these data still need to be supported by a more precise analysis of reporter expression domains at different developmental stages.

To conclude, our results support a scenario for the emergence of the vertebrate lateral plate mesoderm and cranial/pharyngeal mesoderm through the segregation of pre-existing cell populations (homologous to amphioxus ventral part of the somites, first pair and posterior, respectively), which, by becoming partly independent from the somites, could evolve new structures in the trunk and in the head (Fig. 7). We also provide new evidence of the existence of a prechordal plate-like territory in amphioxus and give insights into how the appearance of vertebrate head muscles developing from the pre-chordal plate and cranial/pharyngeal mesoderm might have been achieved through the co-option of *Pitx2* and *Tbx1* for the control of myogenesis.

## Methods
### Ethics approval
All the experiments were performed following the Directive 2010/63/ EU of the European parliament and of the council of 22 September 2010 on the protection of animals used for scientific purposes. Ripe adults from the Mediterranean invertebrate amphioxus species (*B. lanceolatum*) were collected at the Racou beach near Argelès-sur-Mer, France, (latitude 42° 32′ 53′′ N and longitude 3° 3′ 27′′ E) with specific permission from the Prefect of Region Provence Alpes Côte d'Azur. Zebrafish embryos were obtained from AB and Tübingen strains, and

manipulated following protocols approved by the Ethics Committee of the Andalusia Government and the national and European regulation established.

## Cell suspension preparation

Adult amphioxus (*Branchiostoma lanceolatum*) were collected at the Racou beach near Argelès-sur-Mer, France. Gametes were obtained by heat stimulation as previously described in ref. 99. Embryos (~100) at 21 hours post-fertilization (hpf, at 19 °C) were washed 2 times in Ca2 + / Mg2 + -free and EDTA-free artificial seawater (CMFSW: 9 mM KCl, 449 mM NaCl, 33 mM Na2SO4, 2,15 mM NaHCO3, 10 mM Tris-HCl). CMFSW was replaced by CMFSW with Liberase TM at 250 µg/mL (Roche, 05401119001). Cells were then dissociated by a series of pipetting and vortexing during 25 minutes at room temperature. The reaction was stopped by the addition of 1/10th volume of 500 mM EDTA. The cell suspension was centrifuged at max speed for 1 min. The pellet was resuspended in CMFSW containing Calcein violet (Invitrogen™, 65085439) and Propidium iodide (PI, 1 µg/mL, Invitrogen™, P3566).

## MARS-seq

Live single cells were selected using a FACSAria II cell sorter. To this end, we sorted only Calcein positive/PI negative cells, and doublet/ multiplet exclusion was performed using FSC-W versus FSC-H. Cells were distributed into 384-wells capture plates containing 2 µl of lysis solution: 0.2% Triton and RNase inhibitors plus barcoded poly(T) reverse-transcription (RT) primers for single-cell RNA-seq. Single-cell libraries were prepared using MARS-seq[29]. First, using a Bravo automated liquid handling platform (Agilent), mRNA was converted into cDNA with an oligo containing both the unique molecule identifiers (UMIs) and cell barcodes. 0.15% PEG8000 was added to the RT reaction to increase efficiency of cDNA capture. Unused oligonucleotides were removed by Exonuclease I treatment. cDNAs were pooled (each pool representing the original 384-wells of a MARS-seq plate) and linearly amplified using T7 in vitro transcription (IVT) and the resulting RNA was fragmented and ligated to an oligo containing the pool barcode and Illumina sequences, using T4 ssDNA:RNA ligase. Finally, RNA was reverse transcribed into DNA and PCR amplified. The size distribution and concentration of the resulting libraries were calculated using a Tapestation (Agilent) and Qubit (Invitrogen). scRNA-seq libraries were pooled at equimolar concentration and sequenced to saturation (median 6 reads/UMI) on an Illumina NextSeq 500 sequencer and using high-output 75 cycles v2.5 kits (Illumina), obtaining 483 M reads in total.

To quantify single-cell gene expression, MARS-seq reads were first mapped onto *Branchiostoma lanceolatum* genome (GCA_927797965.1, annotation version 3) using STAR v2.7.3[100] (with parameters: −out-FilterMultimapNmax 20 −outFilterMismatchNmax 8) and associated with exonic intervals. Mapped reads were further processed and filtered as previously described[29]. Briefly, UMI filtering includes two components, one eliminating spurious UMIs resulting from synthesis and sequencing errors, and the other eliminating artefacts involving unlikely IVT product distributions that are likely a consequence of second strand synthesis or IVT errors. The minimum FDR q-value required for filtering in this study was 0.02.

## Single cell transcriptome clustering

We used Metacell 0.37[30] to select gene features and construct high-granularity cell clusters (metacells), which were further annotated into cell types (see below). First, we selected informative genes using the *mcell_gset_filter_multi* function in the *metacell* R library, including genes fulfilling these criteria: a total gene UMI count > 30 and >2 UMI in at least three cells, a size correlation threshold of -0.1, and a normalized niche score threshold of 0.01. This resulted in the selection of 844 genes to be used for downstream clustering. Second, we used these genes to build a *K*-nearest neighbors cell graph with *K* = 100

(*mcell_add_cgraph_from_mat_bknn* function), which was the basis to define metacells with an additional *K*-nearest neighbor procedure (*mcell_coclust_from_graph_resamp* and *mcell_mc_from_coclust_balanced* functions) using *K* = 30, minimum metacell size of 15 cells, and 1000 iterations of bootstrap resampling (at 75% of the cells); and a threshold α = 2 to remove edges with low co-clustering weights. Third, we removed one metacell which exhibited low transcriptomic information (> 50 cells with a median UMI/cell <500). This resulted in 176 metacell clusters, which were annotated to known cell types (Supplementary Data 4) based on the expression level of known markers (Supplementary Fig. 1). This same metacell clustering procedure has been applied to each of the neurula stages of *B. floridae*[48].

We recorded gene expression in cell clusters (metacells or cell types) by computing a regularized geometric mean within each cluster and dividing this value by the median across clusters. This normalized gene expression can be interpreted as an expression fold change (FC) for a given metacell or cell type.

Two-dimensional projection of the metacells were created using a force-directed layout based on the metacell co-clustering graph (*mcell_mc2d_force_knn* function).

Gene expression profiles across cell clusters were visualized with heatmaps, using the *ComplexHeatmap* 2.10.0 R library[101]. Cell cluster ordering was fixed according to annotated cell types; and gene order was determined using the highest FC value per cluster. Genes were selected based on minimum differential expression per metacell/cell type, with a maximum number of markers per clusters selected in each case (the actual thresholds used in each heatmap are specified in the corresponding figure legends).

Finally, we selected cells belonging to the endoderm, neural and somitic metacells (Supplementary Data 4), and reclustered them using the same *metacell*-based approach as described for the whole dataset (except that in this case we allowed for smaller metacells, with 10 cells; (Supplementary Data 4). The two-dimensional arrangement of the resulting metacells was curated based on the expression of cell type-specific known markers of various cell subtypes (Supplementary Figs. 5–7).

## ATAC-seq library preparation

For ATAC-seq library construction, 25 embryos at the 21 hpf (19 °C) were transferred in a 1.5 ml tube, in four replicates. We then followed the method described in ref. 102. After a one minute centrifugation at 13 000 rpm, seawater was carefully removed. 50 µl of cold lysis buffer (10 mM Tris-HCl, pH 7.4, 10 mM NaCl, 3 mM MgCl2, 0.1% Igepal) were added and cells were lysed by gentle pipetting. While 25 µl of the lysate was centrifuged at 500 g for 10 minutes at 4 °C, the other 25 µl were used to count nuclei after DNA labeling with DAPI and around 50 000 nuclei were used per transposition reaction. The supernatant was removed, the nuclei resuspended in the reaction mix (25 µl 2x TD buffer (Illumina), 2,5 µL Tn5 transposase (Illumina), 22,5 µL nuclease free H2O) and incubated at 37°C for 30 minutes. Following transposition, 3 µl of 3 M AcoNa (pH5.3) were added to the reaction to adjust the pH, and the DNA was purified using the MinElute PCR purification Kit (Qiagen), following the manufacturer's instructions. The transposed DNA was eluted in 10 µL elution buffer preheated at 37 °C. To amplify the library, the following components were combined: 10 µL of transposed DNA, 10 µL of nuclease free H2O, 2,5 µL Nextera PCR primer 1 (25 µM), 2.5 µL Nextera PCR primer 2 (25 µM) and 25 µL NEB-Next® high-fidelity 2x PCR master mix (NEB). We used the following conditions for PCR amplification: 72 °C for 5 minutes, 98 °C for 30 seconds, followed by 13 cycles at 98 °C for 10 seconds, 63 °C for 30 seconds and 72 °C for 1 minute. Following PCR amplification, 3 µl of 3 M AcoNa (pH5.3) were added to the reaction to adjust the pH, and the library was purified using the MinElute PCR purification Kit (Qiagen), following the manufacturer's instructions using 20 µL of elution buffer preheated at 37 °C.

## Analysis of neurula regulatory regions

We used the ATAC-seq data from the 21 hpf embryo to build a catalog of neurula regulatory regions. For comparison, we also used previously published[26] ATAC-seq libraries of 15 hpf and 36 hpf embryos (the closest developmental timepoints available in that study; NCBI SRA accession numbers SRR6245277 to SRR6245279), as well as H3K4me3 ChIP-seq libraries from these same timepoints (SRA accession numbers SRR6245317 to SRR6245320).

The ATAC-seq libraries corresponding to the 15, 21 and 36 hpf embryos were mapped separately to the *B. lanceolatum* genome using *bwa* 0.7.17 (*mem* algorithm[103]). The resulting BAM files were (i) filtered using *alignmentSieve* (from the *deeptools* 3.5.1 package[104]) to exclude weak alignments MAPQ > 30), (ii) corrected to shift the left and right ends of reads, to account for ATAC mapping biases (+ 4/− 5 bp in the positive and negative strands, using the --*ATACshift* flag in *alignmentSieve*), and (iii) filtered to only include nucleosome-free alignments (--*maxFragmentLength 120* with *alignmentSieve*). Duplicated reads were marked with *biobambam2* 2.0.87[105], coordinate-sorted, and removed to produce filtered BAM files. Then, we concatenated the BAM files stage-wise. Normalized coverage for each stage was reported as bins per million mapped reads (BPM), calculated using the *bamCoverage* tool in *deeptools*. The ChIP-seq libraries for 15 and 36 hpf were processed in the same way (except for the ATAC mapping bias correction step and the filtering of nucleosome-free alignments).

For the 21 hpf ATAC-seq experiment, we used *MACS2* 2.2.7.1[106] to identify regulatory elements with the *callpeak* utility, starting from the nucleosome-free filtered BAM file, with the following options: (i) an effective genome size equal to the ungapped amphioxus genome length, (ii) keeping duplicates from different libraries (--*keep-dup all* flag), (iii) retaining peaks with a *q*-value < 0.01, (iv) enabling multiple summit detection (--*call-summits* flag), and (v) disabling the modeling of peak extension for ChIP-seq libraries (--*nomodel* flag).

We then assigned the *MACS2*-predicted regulatory elements to their proximal genes, based on their distance to each gene's transcription start site (TSS). Specifically, we selected well-supported *MACS2* regulatory elements (*q*-value < 1 × 10$^{-6}$), standardized their lengths to 250 bp (125 bp to each side of the predicted peak summit), and assigned each peak to nearby genes based on distance to their TSS (excluding genes further away than 20 kbp, and genes located beyond a more proximal gene). Peaks overlapping the promoter region of a particular gene (defined based on TSS coordinates +/− 50/200 bp or coincidence with H3K4me3 ChIP-seq peaks for the 15 and 36 hpf datasets) were not assigned to any other gene. The peak sets were reduced to non-overlapping sets to avoid redundant regions. These genome coordinate operations were done using the *GenomicRanges* 1.46 and *IRanges* 2.28 packages in *R*[107]. We used these gene-regulatory element assignments to define lists of cell type-specific regulatory elements, based on the expression specificity of each gene (expression fold change ≥ 1.5 in a given cell type). In parallel, we also defined a set of background regulatory regions for each cell type (consistent of regulatory regions linked to non-overexpressed genes, at fold change ≤ 1). In total, we assigned 51,028 regulatory regions (ATAC peaks) to 19,069 genes (out of 27,102), with a median of 2 peaks per gene.

We used the cell type-specific sets of active regulatory elements (and their corresponding background sets) to identify motifs de novo using the *findMotifsGenome.pl* utility in *homer* 4.11[108]. Specifically, we set a constant peak size of 250 bp and attempted to identify motifs for each cell type, using *k*-mers of length 8, 10, 12, and 14; and tolerating up to four mismatches in the global optimization step.

In order to build a final motif collection for amphioxus, we concatenated the cell type-specific de novo motifs with known TF binding motifs from the CIS-BP database (as available the 3rd of March, 2023)[109]. Specifically, we used 3547 experimentally determined motifs (with SELEX or PBMs), corresponding to vertebrate or tunicate species (*Homo sapiens*, *Mus musculus*, *Xenopus tropicalis*, *Xenopus laevis*, *Danio*

*rerio*, *Tetraodon nigroviridis*, *Meleagris gallopavo*, *Gallus gallus*, *Anolis carolinensis*, *Takifugu rubripes*, *Ciona intestinalis*, and *Oikopleura dioica*). We reduced the redundancy of this extensive de novo + known motif collection based on motif-motif sequence similarity, as follows: (i) we removed motifs with *homer* enrichment *p*-values < 1 × 10$^{-9}$; (ii) we retained with high contiguous information content (IC), defined as having IC ≥ 0.5 for at least four consecutive bases or IC ≥ 0.5 for two or more blocks of at least three bases; (iv) for each of the remaining motifs, we measured their pairwise sequence similarity by calculating the weighted Pearson correlation coefficient of the position probability matrices of each motif, using the *merge_similar* function in the *universalmotif* 1.12.4[110] R library with a similarity threshold = 0.95 for hierarchical clustering and a minimum overlap of 6 bp between two motifs in the motif alignment step. Finally, we selected the best motif per cluster based on its IC (highest). This resulted in a final, non-redundant collection of 1,595 motifs.

Then, we calculated the enrichment of each motif among the sets of regulatory regions specific to each cell type. To that end, we used the *calcBinnedMotifEnrR* function in the *monalisa* 1.0 R library[111] to count motif occurrences in three sets of regulatory regions (bins) defined based on the expression levels of their associated genes: highly cell type-specific genes (FC ≥ 1.5), mildly cell type-specific genes (FC ≥ 1.1 and <1.5), and non-cell type-specific genes (FC < 1). Motif occurrences were defined as motif alignments with scores above 80% of that motif's maximum alignment score (defined from the corresponding position weight matrices). Motif enrichment in each bin was then calculated using the fold change of occurrence relative to randomly sampled genomic regions (matched by GC content and length, using twice as many regions for background as for the foreground), and its significance assessed using a binomial test followed by Benjamini-Hochberg *p*-value adjustment. We retained the fold change and *p*-values for th set of highly cell type-specific regulatory regions (i.e. from genes with FC ≥ 1.5) for further analysis (Fig. 3 and Supplementary Data 1).

Finally, we scanned the *B. lanceolatum* genome to identify discrete occurrences of each of the 1595 motifs across the 51,028 *MACS2*-defined regulatory regions. We used the *findMotifHits* function in *monalisa*. In order to define bona fide motif alignments, we calculated an empirical *p*-value for each motif alignment (only best alignment per regulatory region) based on the rank of its alignment score when compared to a background distribution of randomly sampled genomic regions of similar sequence composition (only best alignment score per random background bin). Specifically, we divided the foreground regions into 10 equal-size sets based on their GC content, and matched each set with random genomic background sequences (not in the foreground) of similar GC content (same category) and equal length (set to 250 bp). These motif alignments were used to identify enhancer-specific motifs in Fig. 5 (complete list in Supplementary Data 2).

## Cross-species cell type comparison

We used SAMap 1.0.2[112] to evaluate the similarity between *B. lanceolatum* cell types and the previously published developmental single-cell transcriptomes of *Danio rerio*[51] (reference gene set in original study: GRCz10 v1), *Xenopus tropicalis*[50] (reference gene set in original study: Xenbase version 9.0), *Ciona intestinalis*[49] (reference gene set in original study: KH2012 from the Ghost Database (http://ghost.zool.kyoto-u.ac.jp/download_kh.html), and *Mus musculus*[52].

For each query species, we used the UMI tables corresponding to the timepoints closest to the *B. lanceolatum* 21hpf developmental stage (12 in total): 14 hpf, 18 hpf and 24 hpf for *D. rerio* (GEO accession: GSE112294); S14, S16, S18, S20 and S22 for *X. tropicalis* (GSE113074); the initial, early, middle and late tailbud stages for *C. intestinalis* (GSE131155) and the stages E7.00, E7.25, E7.50, E7.75, E8.00, E8.25 and E8.50 for *M. musculus*. For *C. intestinalis*, we used the cell type

annotations used in the original paper. For *D. rerio* and *X. tropicalis*, we used the consensus cell annotations employed by Tarashansky et al.[112].

To run SAMap, we first created a database of pairwise alignments with *blastp* 2.5.0 (comparing *B. lanceolatum* peptides to each query species separately; in the case of *Danio rerio* we used *blastx/tblastn* instead of *blastp* as the original gene set[51] was only available as un-translated transcripts). Second, we used the cell-level UMI counts of each gene to calculate the SAMap mapping scores for each pair of cell types (between *B. lanceolatum* and each of the 12 query developmental datasets in other species), using all cells within each cluster for score calculation. SAMap identifies the best cross-species markers from the BLAST-based homology graph using an iterative procedure based on gene-gene expression correlation and cross-species embedding of the transcriptomic manifolds[112].

Finally, we identified shared marker genes between cell types of *B. lanceolatum* and the query chordate species by identifying sets of cell type-overexpressed genes with the *scanpy* 1.9.3[113] *rank_genes_groups* function to calculate cell type-level fold change values and over-expression significance (Wilcoxon rank-sum tests followed by BH *p*-value adjustment). For each species, cell type-specific genes were then determined based on fold change and overexpression significance (at adjusted $p < 0.05$ and $FC \geq 1$).

For cross-species comparisons, genes were linked based on shared orthology group membership. Orthology groups between genes of the the four species were determined using *Broccoli* 1.1[114] (using predicted peptides as input; disabling the *k*-mer clustering step; using up to 10 hits per species for maximum-likelihood phylogenetic tree calculations. If available, we gave preference to phylogeny-derived annotation of orthology groups (see below) over the *Broccoli* orthology assignments. The orthology group assignments of all genes are available in Supplementary Data 3.

We also performed a more detailed analysis of shared TFs between amphioxus and the other three chordates, selecting cell type-specific amphioxus TFs ($p < 0.05$ and $FC \geq 1.25$; see below details on TF annotation) and evaluating whether their orthologs in chordates were also over-expressed in cell types homologous to the amphioxus endoderm (in this case, it was compared to endodermal tissues in the other chordates), endostyle (to other endodermal tissues), muscular somites (to vertebrate skeletal muscle and tunicate muscle/heart), somites (to vertebrate presomitic mesoderm or tunicate muscle/heart), notochord (to other notochordal tissues) hypothalamus and neurons (each of which was compared to vertebrate neurons, hindbrain, forebrain/midbrain, notoplate and neuroendocrine cells; and to the tunicate nervous system), and the anterior and posterior epidermis (each compared to epidermal progenitors, ionocytes, small secretory epidermal cells, goblet cells, and hatching gland).

## Gene family annotation

We ran gene phylogenies to refine the orthology assignments of TF gene families. We used translated peptide sequences from 32 metazoan (longest isoforms per gene, Supplementary Data 3, which were scanned using *hmmsearch* (*HMMER* 3.3.2[115]) to identify hits of TF-specific HMM profiles (from Pfam 33.0[116]) representing their corresponding DNA-binding regions. For each gene family, the collection of homologous proteins was aligned to itself using *diamond blastp* v0.9.36[117] and clustered into low-granularity homology groups using the Markov Cluster Algorithm *MCL* v14.137[118] (using alignment bit-scores as weights, and a gene family-specific inflation parameter; Supplementary Data 3). Then, each homology group was aligned using *mafft* 7.475[119] (E-INS-i mode, up to 10,000 refinement iterations). The alignments were trimmed with *clipkit* 1.1.3[120] (*kpic-gappy* mode and a gap threshold = 0.7) and used to build phylogenetic trees with *IQ-TREE* v2.1[121] (running each tree for up to 10,000 iterations until convergence threshold of 0.999 is met for 200 generations; the best-fitting evolutionray model was selected with *ModelFinder*[122]; statistical supports

were obtained using the UFBoot procedure with 1,000 iterations[123]). Outlier genes were removed from each tree using *treeshrink* v1.3.363 (gene-wise mode using the centroid rooting algorithm; scaling factors set to $a = 10$ and $b = 1$); and the trees were recalculated if necessary if any outgroup needed to be removed. Finally, we used *Possvm* 1.1[124] to identify orthology groups from each gene tree (with up to 10 steps of iterative gene tree rooting), and annotated the orthogroups and the *B. lanceolatum* TFs with reference human gene names.

For genes used to assign metacells to known amphioxus embryonic territories and named in the manuscript, we either used the previously published amphioxus gene names when they exist, or a name based on fine orthology analysis. Amino acid sequences from *B. lanceolatum* were used to search Genbank for putative homologs by *blastp*. Sequences were aligned using *ClustalX*[125]. Alignments were manually corrected in *SeaView*[126]. Maximum Likelihood phylogenetic trees were reconstructed using *IQ-TREE* v2.1[121] with default parameters (fast bootstraping and automatic best model search). Genes with no clear orthology signal were named based on the presence of known protein domains.

## In situ hybridization and immunostaining

DIG labeled probes were synthesized from fragments cloned into pBKS, or from PCR amplified DNA fragments purchased at Integrated DNA Technologies, Inc (IDT), using the appropriate RNA polymerase (T7, T3 or SP6; Roche, RPOLT7-RO, RPOLT3-RO, RPOLSP6-RO) and the DIG-labeling Mix (Roche, 11277065910). Amphioxus embryos at 21 hpf (19 °C) were fixed in paraformaldehyde (PFA) 4% in MOPS buffer, dehydrated in 70% ethanol and kept at -20 °C. Colorimetric In situ hybridization was undertaken as previously described in[36]. HCR in situ hybridization was performed as described in[127]. Ten embryos were used for each experiment. The probes were designed using the following generator (https://github.com/rwnull/insitu_probe_generator) as published in[128] and obtained from IDT, and HCR amplifiers (B1-Alexa647 for *Brachyury2*, *Gata1/2/3*, and *Tcf21/Msc*; B2-Alexa488 for *Dmbx*, *MLC-alk* and *Ripply*; B3-Alexa594 for *Alx* and *Fgfrl*) were obtained from Molecular Instruments, Inc. Immunostaining after HCR was undertaken as described in[14]. We used anti-laminin antibody produced in rabbit at 1:50 dilution (Sigma, L9393), and the Goat anti-Rabbit IgG (H + L) Cross-Adsorbed Secondary Antibody, Alexa Fluor" 680 at 1:500 dilution (Invitrogen™, A-21076). For imaging, embryos were embedded into ProLong™ Diamond Antifade Mountant with DAPI (Invitrogen™, P36961) and confocal stacks were acquired on whole embryos on a Leica SP8 Confocal microscope using a 40X oil-immersion objective. Images were processed using ImageJ (Fiji) and IMARIS v9.7 (Bitplane). Zebrafish embryos were raised until the desired stage, fixed in 4% PFA and dehydrated in methanol. HCR technique was performed as previously described in[129] and Molecular Instrument's web page (https://www.molecularinstruments.com/hcr-rnafish-protocols). 20 embryos were used for each HCR experiment. Oligo pools used as probes were designed using Easy_HCR[130] and obtained from IDT. HCR amplifiers (B5-Alexa Fluor-647 for *Prrx1a* and B1-Alexa Fluor-546 for *GFP*) were obtained from Molecular Instruments, Inc. The accession numbers/sequences used for probe synthesis or as HCR probes are given in Supplementary Data 5.

## Zebrafish transgenesis

The putative regulatory regions were cloned after PCR amplification on genomic DNA in the PCR8/GW/TOPO vector (Life Technologies). Using Gateway technology (Life Technologies), the inserts were then shuttled into an enhancer detection vector composed of a *gata2* minimal promoter, an enhanced GFP reporter gene, and a strong midbrain enhancer (z48) that works as an internal control for trans-genesis in zebrafish[131]. Transgenic embryos were generated using the Tol2 transposase system[132]. Briefly, 1-cell stage embryos were injected with 2 nl of a mix containing 25 ng/μL of Tol2 transposase mRNA,

20 ng/µL of purified vector, and 0.05% of phenol red. Zebrafish embryos were obtained from AB and Tübingen strains at the fish facility of Centro Andaluz de Biología del Desarrollo (Seville, Spain). Each crossing implied 10 females and 10 males aged 4 months. The sex of the embryos was not determined. 300 embryos were injected at one-cell stage for the generation of each transgenic line. We then raised the embryos that exhibited a strong GFP expression in the midbrain (approximately a 15-20% of them), driven by the z48 enhancer. When these animals (F0) reached adulthood (4 months), they were outcrossed with wild type animals in order to identify founders and generate stable transgenic lines (F1). Around 30% of the selected animals resulted to be founders, generating a stable transgenic offspring. An element was considered negative when at least three founders transmitted GFP expression only in the midbrain. On the contrary, when other domains besides the midbrain were observed in the offspring of at least three independent founders, and presented no (or very subtle) variation between them, this element was considered positive. From all the constructs injected and screened for F1, 3 showed 3 founders with similar expression (see Supplementary Data 6). The other animals screened showed the expression of the internal control in the midbrain, but not a consistent and specific expression in other domains in 3 independent founders. Embryos were raised until the desired stage, visualized under an Olympus SZX16 fluorescence stereoscope and photographed with an Olympus DP71 camera or fixed in PFA for in situ hybridization. The transgenic lines are no longer available.

### Constitutive repressor mRNA injection

Constitutive repressor form of Gata1/2/3 was created by fusing the coding sequence of the repressor domain of the engrailed protein[133] to the N-terminal side of the full-length sequence of Gata1/2/3. The sequence (see Supplementary Data 5) was inserted into pCS2+. The vector was linearized and in vitro transcription was performed using the mMESSAGE mMACHINE® SP6 Transcription Kit (Invitrogen™, AM1340). Microinjections were carried out as described in[134] with some minor changes. Briefly, eggs were fertilized and dechorionated by pipetting. They were placed in an Petri dish coated with agarose and microinjected with a mix containing 1.5 µg of Gata1/2/3-Engrailed mRNA, 0,5 µg of mCherry mRNA, 18% of glycerol and 18% of Fast Green. 10 hours after fertilization, the embryos showing red fluorescence were placed in a new Petri dish without coating. The embryos were then fixed when desired for subsequent in situ hybridization experiments.

### Reporting summary

Further information on research design is available in the Nature Portfolio Reporting Summary linked to this article.

### Data availability

The *B. lanceolatum* sequencing data generated in this study have been deposited in the GEO database under accession code GSE255742, and the corresponding processed gene expression data are available in this same database and the Source Data file. The gene expression data from other chordates, generated in previous studies, are available in the following databases: for *Ciona intestinalis* GEO database GSE131155; *M. musculus*, *D. rerio* and *X. tropicalis*, TOME database (http://tome.gs.washington.edu/); *B. floridae*, the publication-specific database (https://lifeomics.shinyapps.io/shinyappmulti/). Accession numbers of sequences used for in situ hybridization probe synthesis are given in Supplementary Data 5. Source data are provided with this paper.

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

## Acknowledgements

We would like to thank Jose Luis Ferran for advices on the revision. This work benefited from access to the Observatoire Océanologique de Banyuls-sur-Mer, an EMBRC-France and EMBRC-ERIC site. Embryo imaging experiments were undertaken using the material of the BIOPIC platform. The laboratory of H.E. and S.B. was supported by the CNRS, and by the "Agence Nationale de la Recheche" under the grants ANR-19-CE13-0011 to H.E. and ANR-21-CE13-0034 to S.B. Research in A.S-P. group was supported by the European Research Council (ERC-StG 851647 to A.S-P.) and the Spanish Ministry of Science and Innovation (PID2021-124757NB-I00 to A.S-P.). X.G-B. is supported by the European Union's H2020 research and innovation program under Marie Sklodowska-Curie grant agreement 101031767 to X.G-B. A.E. was supported by FPI PhD fellowships from the Spanish Ministry of Science and Innovation. J.J.T. was supported by the Spanish Ministerio de Economía y Competitividad (grant PID2019-103921GB-I00 and PID2022-141288NB-I00 to J.T.). M.I. laboratory research has been funded by the European Research Council (ERC) under the European Union's Horizon 2020 research and innovation program (ERCCoG-LS2-101002275 to M.I.), by the Spanish Ministry of Economy and Competitiveness (PID2020-115040GB-I00 to M.I.) and by the 'Centro de Excelencia Severo Ochoa 2013-2017'(SEV-2012-0208 to M. I.).

## Author contributions

Conceptualization of this study was done by J.L. G-Z., M.I., S.B., A.S.P.and H.E.; the study was carried out by X.G.B., L.S., L.M., A.S, A.N., A.E., S.N., O.F., M.I., S.B., A.S.P. and H.E.; writing of the original draft was done by X.G.B., S.B., A.S.P. and H.E.; funding was acquired by A.S.B., J.T., M.I., S.B., A.S.P. and H.E.; this study was supervised by J.L. G-Z, J.T., M.I., S.B., A.S.P. and H.E.

## Competing interests

The authors declare no competing interests.
