## [Peer Review File · Nature Communications]

An amphioxus neurula stage cell atlas supports a complex scenario for the emergence of vertebrate head mesodermREVIEWER COMMENTS

Reviewer #1 (Remarks to the Author):

In the submitted manuscript, Grau-Bové, Subirana, et al. describe a single cell-level analysis of mesodermal lineage composition in amphioxus towards investigating the evolutionary trajectory of head mesoderm. This is an increasingly studied topic in the field, and comparative work as described by the authors has made a substantial contribution to our contemporary understanding of how distinct mesodermal lineages contribute to rostral structures in chordates/vertebrates. The authors describe a single cell transcriptome-level analysis of different lineages in amphioxus that they link by comparative means to mesodermal lineages in vertebrates, in particular lateral plate mesoderm, pharyngeal mesoderm, and prechordal plate populations. The authors further venture out to test gene-regulatory elements of major cluster-defining amphioxus genes using zebrafish reporter assays.

Overall, the authors tackle an exciting line of research and contribute a dataset of potential wide interest to the field. However, beyond the laudable single-cell and ATAC-seq data, the current manuscript version provides at times little beyond the -omics data in terms of conceptual or mechanistic validation. Several major points do require addressing before publication of the manuscript in a suitable journal.

Throughout the manuscript, the results of the authors would benefit from more context to already established findings in the field, e.g. prior work linking ventral somite territories as LPM equivalent, established expression domains as LPM-like patterns in amphioxus, etc.. While validation of scRNA-seq findings is always challenging, and the authors show beautiful examples of expression pattern validations, these are comparatively limited in the currently presented form. Further, the presented zebrafish reporter assays require revisiting considering the current state-of-the-art and would greatly benefit from equivalent tests (albeit challenging) in amphioxus.

*** Major points:

1) Gata1/2/3, Tbx1/10, and Pitx enhancers are presented by testing in zebrafish (see also below – reporter testing in amphioxus to validate that their activity matches endogenous expression would significantly strengthen the manuscript. Such tests (as well-represented in previous literature) are critical for making the claim about the relationship with vertebrate tissues that show zebrafish enhancer activity.

2) Zebrafish reporter tests: these experiments have, in the currently presented form, several issues that interfere with the proper interpretation and validity of the presented conclusions. The level of provided detail also greatly contrasts to the impressive analyses of amphioxus expression patterns provided (e.g. Fig 4). The confirmation in germline-transmitted lines is laudable. Nonetheless, the zebrafish field has made tremendous progress in regulatory element testing that the authors should use to revisit their experiments to bring them up to current standards. Concretely:

2a) The chosen transgenesis marker obscures major parts of the anticipated expression patterns and interferes with the regulatory element tests significantly. Further, there appears to be low level fluorescence posterior to this marker and in the trunk CNS. This is not described in the text (ectopic expression?). Overall, several much more suitable and validated transgenesis markers have been established in the zebrafish field (myl7, a-crystallin, etc.) that would greatly help with these experiments and should be used when revisiting the enhancer testing.

2b) The authors claim several details about expression patterns, yet the provided images are low-res, small, and have the transgenesis marker obscuring major parts of the anatomy (see also above). Higher resolution pictures are needed to identify distinct structures and to solidify the made claims. For instance, EGFP-positive “head mesoderm” should be further defined by structure – the cardiopharyngeal/LPM progenitors form various structures already at the described timepoints.

2c) Images, stats of all F0 enhancer testing data should be shown in supplemental. As such, also negative data is still critical to show. Information on how many founders, variations in expression patterns per stable line, etc. is all missing and critical to interpreting the experiments fully.

2d) No comparisons are provided to previously established zebrafish reporters for gata genes and Tbx1 (e.g., Meng et al., 1999; Yang et al., 2007; Dobrzycki et al., 2020; Felker et al., 2018; Yuang et al., 2018; Song et al., 2022), for instance co-injection of the authors' amphioxus-based reporters (with complementary fluorophores) into established zebrafish strains (most of which should be accessible to the authors). Direct comparison would greatly strengthen the authors' claims. Additionally or alternatively, the authors can use pre-established zebrafish reporter plasmids from studies above or others, and perform the same injection-based tests to compare expression patterns and dynamics to their newly identified enhancers as minimal comparative means.

3) While the ISH panel validation provides some information on the potential role of these genes in amphioxus (based on position/location of expression), loss-of-function experiments on several key genes would significantly strengthen the manuscript. If the authors have the ability to do so, CRISPR-Cas9-based targeting of Gata1/2/3, Tbx1/10, Pitx would greatly augment the impact of the manuscript.

4) Throughout, the authors focus on a selective set of previous literature (including their own work) to underline their conceptual point. However, especially comparative work on lateral plate mesoderm origins has made tremendous progress in recent years. Why the authors do not cite and contextualize prior work (some even prior to their previous work) is unclear, also as these prior results very much validate and augment the validity of the authors' new work. The authors are encouraged to revisit these prior manuscripts and contextualize their work in the light of these prior findings. For example:

- Onimaru et al., 2011 (in *Dev. Bio.*) established the first analyses of several genes now re-visited by the authors in their study, including Tbx1/10, as seminal paper for the context of lateral plate equivalents across chordates, as now expanded in the authors manuscript and Fig 6.
- Prummel et al., 2019 (also in *Nature Comms.*) established that a small enhancer of the zebrafish *drl* gene provides a LPM reporter when tested in several vertebrates, and labels lateral plate-equivalent cells in *Ciona* and amphioxus. This reporter clearly shows ventral somites as lateral plate mesoderm equivalent and discusses this as such, as now transcriptionally confirmed by the authors' work. Further, this prior study implicates Eomes as one of the transcription factors acting on the reporter, and the authors now also found Eomes in the corresponding LPM/ventral somite domain in their scRNA-seq.
- Yong et al., 2021 (in *Front Cell Dev Biol.*) performed detailed analyses of the lateral wall in amphioxus, concluding equivalence to the vertebrate dermomyotome (dorsal) and to the lateral plate mesoderm (ventral). This prior work also provides detailed outlines of expression patterns of several genes the authors now found in their scRNA-seq, greatly supporting their findings.

5) The authors perform several highly interesting cross-species comparisons using their amphioxus scRNA-seq dataset. However, they opted to use broad, pan-embryonic datasets for zebrafish and clawed frogs that grossly under-represent the mesodermal lineages the authors are interested in. The authors should re-visit these cross-comparisons with existing datasets that are more focused. Such comparisons likely also yield more validated, accurate cluster and gene ontology as the mass-catalog datasets used relied on broad cluster definitions for such niche lineages like cardiopharyngeal or lateral mesoderm. Comparing with these datasets would greatly augment and focus the authors' work:

- Nomaru et al., 2021 (*Nat. Comms*) published a mouse-based single-cell dataset of Tbx1-expressing cardiopharyngeal lineages.
- Prummel et al., 2022 (*Nat. Comms*) published a zebrafish lateral plate dataset at the end of gastrulation.
- De Bono et al., 2023 (again *Nat. Comms*) build upon Nomaru et al. and provide further details in their mouse dataset.

*** Minor and text points:

A) The definitions of lateral plate mesoderm, pharyngeal mesoderm, "head" mesoderm, cardiopharyngeal mesoderm, and the likes have become blurred and at times improperly used in the literature. The authors' work provides a potentially powerful view point to remedy this, and the authors are encouraged to discuss how their data supports (or refutes) differences between these different mesoderm "definitions".

B) Line 62: Pharyngeal mesoderm also encompasses lateral plate, not solely paraxial.

C) Line 158: ATAC-seq data feels a little out of place here as it is breaking up multiple scRNA-seq sections.

D) Line 204: Why did the authors chose to describe neural clusters in such detail, seems misaligned from the general goal of the paper?

E) Throughout the text: please clarify whether "newly described" means by your group in this study, otherwise properly cite original literature.

F) Line 301-302, missing word: "The amphioxus orthologues of the latest were also shown to be expressed in the first somite pair".

Reviewer #2 (Remarks to the Author):

Grau-Bové et al. constructed an amphioxus neurula stage cell atlas. In general, I have several comments for improvement.

1. The authors are encouraged to develop an open access website, or upload their data sets to UCSC cell browser to allow users freely explore the gene expression patterns in different clusters.
2. In this study, authors employed MARS-seq for scRNAseq library construction. Have the authors tested other methods, for example 10x chromium or Dropseq, what benefits could MARS-seq particularly provide? Would the discovery/ statement still hold true if more cells were sampled?
3. The authors should state the limitation of this study in the discussion section.
4. In addition to *Ciona intestinalis*, *Danio rerio*, and *Xenopus tropicalis*, authors are suggested to compare amphioxus atlas with that of human or mouse to reveal the conservation and divergence of genetic regulatory networks.
5. The bubble plot in Figure 2a is too crowded.
6. More details are required for cross-species homologs identification.
7. Previous published amphioxus single cell atlas should be discussed.

Reviewer #3 (Remarks to the Author):

The manuscript by Grau-Bové, is a single cell atlas of the neurula stage in the amphioxus *Branchiostoma lanceolatum*. The work exploits publicly available single-cell datasets for comparisons with other species, and bulk-ATAC seq at three different developmental stages (two of which were previously published by the same group) to interrogate the regulatory logic behind the observed cell-specific transcriptional profiles. The authors claim support for the existence of a transient prechordal plate structure in amphioxus, and anterior mesodermal regions homologous to the vertebrate

cranial/pharyngeal and lateral plate mesoderm. The manuscript finishes with the proposal of a stepwise scenario for the segregation and remodelling of the anterior axial mesoderm into prechordal plate, cranial/pharyngeal mesoderm, and lateral plate mesoderm during vertebrate evolution. Co-option of *Pitx2* and *Tbx1/10* to the myogenic program in vertebrates would have been key in this evolutionary transition.

One of the main concerns is the lack of novelty. I cannot find anything in the manuscript that we did not know before. The Escriva lab already proposed that amphioxus might have something similar to a prechordal plate in a nice work that combines photoconvertible reporters and functional assays here: [10.1242/dev.200252](https://doi.org/10.1242/dev.200252). The same lab has also published research on the evolution of the vertebrate head mesoderm and proposed the same evolutionary scenario depicted here (see for example [10.1038/s41559-019-0933-z](https://doi.org/10.1038/s41559-019-0933-z)). Apart from that, this is not the first single-cell genomics dataset published in amphioxus, and I can only define as striking, the lack of acknowledgement to the work of Ma and collaborators, published a year ago (see <https://doi.org/10.1016/j.celrep.2022.110979>). Ma and collaborators report a total of 148875 cells through 9 developmental stages of the amphioxus *Branchioma floridae*, with the correspondent ATAC-seq, that in this case is also single-cell and not in bulk, as presented by Grau-Bové and collaborators. Grau-Bové use the same integration method as Ma, so I find hard to believe they have not read the work. This is not only a matter of acknowledgment but a missed opportunity to understand how single-nucleus compares to single-cell profiling, since Ma and collaborators cover the same developmental stage, N3, with a total of 22253 cells, which is a higher number than the number of cells reported by Grau-Bové at the same stage. Also, a missed opportunity for the authors here, to find support for their own results. Given that this is not the first single-cell dataset produced in amphioxus, at the very least, Grau-Bové and collaborators should have studied how their dataset fits within that previously produced by Ma et al., which is a much more comprehensive study. As it stands, I cannot see what the Grau-Bové dataset adds to what it was already described by Ma and collaborators.

An even major concern is the number of imprecisions throughout the manuscript. The use of metacells is clever, especially if you have sparse data, but it also indicates the dataset lacks granularity. This is not a negative per se, but a limitation that the authors should acknowledge by: i) avoiding fine resolution annotations; ii) or by validating their proposed annotations using orthogonal approaches. This could well be in situ, but not the sort of in situ included in this study, which do not offer single-cell resolution nor co-expression information, more like the sort of in situ the Escriva lab published previously here: [doi:10.1242/dev.200252](https://doi.org/10.1242/dev.200252). Because the authors have not done one of the other (i or ii), the annotations and conclusions are extremely confusing. A few examples of such imprecisions are detail below:

1. Mesoderm annotations do not coincide with the profiles previously published by the Escriva or Yu labs. Works from the two labs showed that mesoderm is highly regionalised anterior-posteriorly, dorso-ventrally and medio-laterally, with specific populations expressing particular combinations of genes. Where are all these cells here? Do the authors confirm? Deny? Or none of the above? The work does not provide further understanding on this matter. Furthermore, the work of the Yu lab is absent from the reference list ([10.3389/fcell.2021.607057](https://doi.org/10.3389/fcell.2021.607057)).
2. How can netrin be localised to metacell 9 only, in the posterior neural tube, when it is expressed along the entire neural tube? See Shimeld, 2000. Netrin must be also in 14, 8, 20 and 21, probably in 12 too. This is surely a phrasing problem. It might be helpful to have the top differentially expressed genes per metacell, to see the different combinations. Overexpression is not always differential expression. As it is written it sounds like a guessing based on the expression of individual genes. The authors submitted a table with the re-clustering, but it does not detail how the profiles have been refined.
3. Another case is that of the epidermis, which of the three classifications are FoxJ or Keratin or none of the above? This does not allow relating this annotation to the clusters defined by Ma and collaborators, and the works published previously on epidermis gene expression. It is very confusing.
4. I do not understand either, the proposal that metacells 4 and 22 correspond to the Hypothalamo-prethalamal primordium. Since Albuixech-Crespo and collaborators decided to rename the amphioxus

forebrain as Hypothalamo-prethalamic primordium at this stage of development (7ss, N3), there has been several publications that demonstrate this does not reflect how the amphioxus brain is structured or develops. The amphioxus brain at this stage has not even started the process of neurogenesis. It has been shown that embryos at the 7ss (N3) do not have hypothalamus nor hypothalamus primordium, that the hypothalamus forms by intercalated growth and addition of new cells at the 7ss and these progenitor cells do not have a recognisable hypothalamic profile until the 11-12ss, when they start expressing *Otp* (see <https://doi.org/10.3389/fnins.2022.812223>). Most of the hypothalamic centres in vertebrates depend on *Otp* expression (see [doi:10.1242/dev.139055](https://doi.org/10.1242/dev.139055)), so the annotations here does not make any sense. *Fezf* is involved in the formation of serotonergic cells of the frontal eye complex in amphioxus, that would correspond to the pre-optic/optic area, which is possibly what the authors see here. *Six3/6* and *Otx*, on the other side, are expressed in very distinct parts of the amphioxus brain, also in vertebrates. There are numerous publications backing this up. These cannot be used on their own to have such refine annotation. *Ctfp1*, has previously been reported at a single-cell resolution in situ (see <https://doi.org/10.1101/2023.03.13.532359>) and it is expressed in two different brain regions, excluding the hypothalamus. This is the reason why Grau-Bové see cells expressing this marker at this stage, because it is not expressed by the hypothalamus. Note that the gene is described as being expressed in the cerebral vesicle as a whole, which is incorrect. The stage analysed by Grau-Bové is too young and the sequencing is too shallow to ascribe these metacells to such particular brain areas. The cross-species comparisons in Figure 2, also show this classification to be arbitrary, as what they call Hypothalamo-prethalamic primordium is hitting everywhere in the anterior nervous system of zebrafish and xenopus. Perhaps annotating these cells as anterior and posterior forebrain would be more appropriate in this case.

5. In the ATAC-seq section Grau-Bové also proposes elements of hypothalamic-specific expression of *Soxb1c*(*Sox2*) and *Pou3fl* (*Pou3f4*), which we know for certain they are not specific of the hypothalamus but more general central nervous system markers, also at the N3 stage, see [doi: 10.7150/ijbs.3.356](https://doi.org/10.7150/ijbs.3.356), and <https://doi.org/10.3390/cells12040614>.

6. I cannot find clarity either about the prechordal plate (metacell 1) is this co-expressing *Gooseoid*, *Dmbx* and *Nodal*, as proposed earlier? Is this *Bra* positive or not? Because the vertebrate prechordal plate is not? And does this express any FGF receptors that could explain the phenotype described by the Escrivá lab before ([10.1242/dev.200252](https://doi.org/10.1242/dev.200252))?

7. The reporter expression in zebrafish should be accompanied by co-detection of GFP with the genes they claim characterise those specific tissues where the reporter is expressed. Otherwise, this only shows these enhancers that are able to drive expression in zebrafish.

There are also some other technical details that are puzzling, 5% of cell recovery and cell types captured in the same proportions as they are in the embryos? It is of course expected to have a low representation of minority cell types, but not following the same ratios. Capture should be random after dissociation. It looks like embryos were partially digested, which would explain why there are so many epidermal cells: they are the first ones to peel off. It is hard to say because there are no quality controls in the paper.

**Please find below our point-by-point answer to the reviewer's comments. For clarity our**
**answers are in blue.**

**Reviewer #1 (Remarks to the Author):**

In the submitted manuscript, Grau-Bové, Subirana, et al. describe a single cell-level analysis of
mesodermal lineage composition in amphioxus towards investigating the evolutionary
trajectory of head mesoderm. This is an increasingly studied topic in the field, and comparative
work as described by the authors has made a substantial contribution to our contemporary
understanding of how distinct mesodermal lineages contribute to rostral structures in
chordates/vertebrates. The authors describe a single cell transcriptome-level analysis of
different lineages in amphioxus that they link by comparative means to mesodermal lineages in
vertebrates, in particular lateral plate mesoderm, pharyngeal mesoderm, and prechordal plate
populations. The authors further venture out to test gene-regulatory elements of major cluster-
defining amphioxus genes using zebrafish reporter assays.

Overall, the authors tackle an exciting line of research and contribute a dataset of potential wide
interest to the field. However, beyond the laudable single-cell and ATAC-seq data, the current
manuscript version provides at times little beyond the -omics data in terms of conceptual or
mechanistic validation. Several major points do require addressing before publication of the
manuscript in a suitable journal.

Throughout the manuscript, the results of the authors would benefit from more context to
already established findings in the field, e.g. prior work linking ventral somite territories as
LPM equivalent, established expression domains as LPM-like patterns in amphioxus, etc..
While validation of scRNA-seq findings is always challenging, and the authors show beautiful
examples of expression pattern validations, these are comparatively limited in the currently
presented form. Further, the presented zebrafish reporter assays require revisiting considering
the current state-of-the-art and would greatly benefit from equivalent tests (albeit challenging)
in amphioxus.

*** Major points:

1) Gata1/2/3, Tbx1/10, and Pitx enhancers are presented by testing in zebrafish (see also below
– reporter testing in amphioxus to validate that their activity matches endogenous expression
would significantly strengthen the manuscript. Such tests (as well-represented in previous
literature) are critical for making the claim about the relationship with vertebrate tissues that
show zebrafish enhancer activity.

We understand the concern of the reviewer. We have indeed undertaken such transgenesis
experiments in amphioxus using the Tol2 system. However, small DNA sequences, such as the
ones tested in the zebrafish in this study, always give mosaic amphioxus transgenics in the F0
generation. In fact, this feature is not exclusive to amphioxus and it is also the case for zebrafish.
Figure R1 shows some embryos injected with two amphioxus constructs corresponding to the
peaks for which we obtained F1 lines in zebrafish (GFP detected by *in situ* hybridization)
showing mosaic GFP expression in the domains where the corresponding genes (Gata1/2/3 and
Pitx) are expressed. Unfortunately, we cannot obtain F1 transgenic lines with *B. lanceolatum*,
as its life cycle duration is 3 years.

**Figure R1.** Expression of *GFP* in embryos injected with the reporter constructs for *Gata1/2/3*,
 and *Pitx* regulatory regions. Two embryos are shown for each series with lateral views (anterior
 to the left and dorsal to the top) and ventral views (anterior to the left). For comparison, embryos
 after ISH for *Gata1/2/3* and *Pitx* are also shown at the bottom. White arrowheads: somites;
 white asterisks: artefactual labelled cells in the presumptive gut lumen.

2) Zebrafish reporter tests: these experiments have, in the currently presented form, several
 issues that interfere with the proper interpretation and validity of the presented conclusions. The
 level of provided detail also greatly contrasts to the impressive analyses of amphioxus
 expression patterns provided (e.g. Fig 4). The confirmation in germline-transmitted lines is
 laudable. Nonetheless, the zebrafish field has made tremendous progress in regulatory element
 testing that the authors should use to revisit their experiments to bring them up to current
 standards. Concretely:

2a) The chosen transgenesis marker obscures major parts of the anticipated expression patterns
 and interferes with the regulatory element tests significantly. Further, there appears to be low
 level fluorescence posterior to this marker and in the trunk CNS. This is not described in the
 text (ectopic expression?). Overall, several much more suitable and validated transgenesis
 markers have been established in the zebrafish field (*myl7*, *a-crystallin*, etc.) that would greatly
 help with these experiments and should be used when revisiting the enhancer testing.

We appreciate the reviewer's suggestion, and apologize for the lack of clarity in the description
 of these experiments. The enhancer z48 used in this reporter is not only serving as a marker of
 transgenesis, but it is used to enhance the expression of other regulatory elements, since it has
 been described to function as a booster, likely opening the chromatin around the transgene
 insertion (Bessa, Tena et al. 2009). It is true that the strong GFP expression driven by the Z48
 element to the midbrain might obscure some of the expected results. To improve and clarify
 our results, we used zebrafish embryos fixed from our previous transgenesis experiments to
 undertake a double labelling using the HCR methodology with probes against GFP and against
 a well-known gene expressed in the head mesoderm and limb buds (*Prrx1a*). We now provide
 high resolution GFP expression data and show a partial overlap with *Prrx1a* expression in
 **Supplementary Fig. 8**, which clearly demonstrate that the amphioxus enhancers drive
 expression in the vertebrate head mesoderm. Unfortunately, the zebrafish lines that we
 generated in this study were produced during the COVID-19 pandemic and due to the
 restrictions linked to the decrease of personnel in the laboratory during that period, we could

not maintain them, so the lines are no longer available. In order to answer the reviewers concern
we could only work with the few fixed embryos we kept as a back-up. Obviously, the generation
of new zebrafish lines will take too long and would too much delay the publication of this work.

2b) The authors claim several details about expression patterns, yet the provided images are
low-res, small, and have the transgenesis marker obscuring major parts of the anatomy (see also
above). Higher resolution pictures are needed to identify distinct structures and to solidify the
made claims. For instance, EGFP-positive “head mesoderm” should be further defined by
structure – the cardiopharyngeal/LPM progenitors form various structures already at the
described timepoints.

As already mentioned above, the transgenic lines are no more available for new pictures to be
taken. However, high resolution images of HCR experiments involving lines with GFP
expression especially difficult to distinguish have been incorporated to the manuscript (new
**Supplementary Fig. 8**) for the enhancer of *Gata1/2/3* and *Tbx1/10*. We now specify in the
manuscript that expression in the head was observed in the mesenchyme of the head and in the
branchial arches for both *Gata1/2/3* and *Tbx1/10* lines (see lines 376 and 409).

2c) Images, stats of all F0 enhancer testing data should be shown in supplemental. As such, also
negative data is still critical to show. Information on how many founders, variations in
expression patterns per stable line, etc. is all missing and critical to interpreting the experiments
fully.

We apologize for this; we have added more details about zebrafish transgenesis methodology
to the manuscript in the Methods section (line 713-725):

*“For all transgenic lines generated in zebrafish, we injected 300-600 embryos and raised those*
*that exhibited a strong GFP expression in the midbrain (approximately a 15-20% of them),*
*driven by the z48 enhancer. When these animals (F0) reached adulthood, they were outcrossed*
*with wild type animals in order to identify founders and generate stable transgenic lines (F1).*
*Around 30% of the selected animals resulted to be founders, generating a stable transgenic*
*offspring. An element was considered negative when at least three founders transmitted GFP*
*expression only in the midbrain. On the contrary, when other domains besides the midbrain*
*were observed in the offspring of at least three independent founders, and presented no (or very*
*subtle) variation between them, this element was considered positive. From all the constructs*
*injected and screened for F1, 3 showed 3 founders with similar expression (see **Supplementary***
***Table 6**). The other animals screened showed the expression of the internal control in the*
*midbrain, but not a consistent and specific expression in other domains in 3 independent*
*founders.”*

We also included a table indicating the variation in expression and the number of founders
(**Supplementary Table 6**). We did not include the pictures of F0 embryos as they show mosaic
transgene expression.

2d) No comparisons are provided to previously established zebrafish reporters for gata genes
and *Tbx1* (e.g., Meng et al., 1999; Yang et al., 2007; Dobrzycki et al., 2020; Felker et al., 2018;
Yuang et al., 2018; Song et al., 2022), for instance co-injection of the authors' amphioxus-based
reporters (with complementary fluorophores) into established zebrafish strains (most of which
should be accessible to the authors). Direct comparison would greatly strengthen the authors'
claims. Additionally or alternatively, the authors can use pre-established zebrafish reporter

plasmids from studies above or others, and perform the same injection-based tests to compare
expression patterns and dynamics to their newly identified enhancers as minimal comparative
means.

We would like to thank the reviewer for this suggestion. However, due to time restrictions, we
had no possibilities to generate new lines, and, as previously explained, the lines used in this
study were produced during the COVID-19 pandemic and are no more available. We hence
decided to use *Prrx1a* endogenous expression as a marker for head mesoderm and limb buds,
tissues that are known to be part of Tbx1 and Gata expression domains in zebrafish, instead of
a direct comparison with zebrafish enhancer lines as suggested by the reviewer.

3) While the ISH panel validation provides some information on the potential role of these
genes in amphioxus (based on position/location of expression), loss-of-function experiments on
several key genes would significantly strengthen the manuscript. If the authors have the ability
to do so, CRISPR-Cas9-based targeting of Gata1/2/3, Tbx1/10, Pitx would greatly augment the
impact of the manuscript.

We fully agree with the reviewer. Concerning *Pitx*, TALEN mutants were produced and
analysis of the mutant phenotype showed that mutation in this gene induce left/right asymmetry
defects but no mesoderm development defects (Xing, Pan et al. 2021). Concerning *Tbx1/10*,
morpholinos injection data were also published showing no somite developmental defects
(Koop, Chen et al. 2014). We have added the following sentence in the manuscript (line 399):
“*Furthermore, functional analysis by TALEN mutagenesis or morpholino injection for Pitx97*
*and Tbx1/1098 respectively, showed that knockout or knockdown of these genes did not alter*
*somite development.*” .

To further test our hypothesis, we analyzed the function of Gata1/2/3 by overexpressing a
constitutive repressor of Gata1/2/3 (Gata1/2/3-engrailed). We show that, at the neurula stage,
the overexpression of the Gata1/2/3 constitutive repressor alters the formation of somites, with
the ventro-lateral region forming epithelial *Alx*-positive spheres instead of starting to elongate
towards the ventral region of the embryo. We show these results in a new figure (Fig. 6)
displaying injected embryos after HCR against *Alx* (gene normally expressed in the ventral part
of the somites) and *MLC-alk* (gene expressed in the myotome part of the somites) together with
anti-laminin immunostaining that allows to visualize the basal lamina, highlighting the
epithelial structures in the embryo.

4) Throughout, the authors focus on a selective set of previous literature (including their own
work) to underline their conceptual point. However, especially comparative work on lateral
plate mesoderm origins has made tremendous progress in recent years. Why the authors do not
cite and contextualize prior work (some even prior to their previous work) is unclear, also as
these prior results very much validate and augment the validity of the authors' new work. The
authors are encouraged to revisit these prior manuscripts and contextualize their work in the
light of these prior findings. For example:

- Onimaru et al., 2011 (in Dev. Bio.) established the first analyses of several genes now re-
visited by the authors in their study, including Tbx1/10, as seminal paper for the context of
lateral plate equivalents across chordates, as now expanded in the authors manuscript and Fig
6.

- Prummel et al., 2019 (also in Nature Comms.) established that a small enhancer of the
zebrafish *drl* gene provides a LPM reporter when tested in several vertebrates, and labels lateral
plate-equivalent cells in *Ciona* and amphioxus. This reporter clearly shows ventral somites as
lateral plate mesoderm equivalent and discusses this as such, as as now transcriptionally
confirmed by the authors' work. Further, this prior study implicates Eomes as one of the
transcription factors acting on the reporter, and the authors now also found Eomes in the
corresponding LPM/ventral somite domain in their scRNA-seq.

- Yong et al., 2021 (in Front Cell Dev Biol.) performed detailed analyses of the lateral wall in
amphioxus, concluding equivalence to the vertebrate dermomyotome (dorsal) and to the lateral
plate mesoderm (ventral). This prior work also provides detailed outlines of expression patterns
of several genes the authors now found in their scRNA-seq, greatly supporting their findings.

We would like to thank the reviewer for this remark and extend our sincere apologies for
inadvertently overlooking numerous prior works that have shaped the evolutionary scenario we
sought to examine in this study. We have added in the introduction a full paragraph
recapitulating these data and we now cite the works mentioned by the reviewer, as well as other
previous findings. We also discuss in the section of somite re-clustering the fact that we do not
find the four compartments described by some authors in amphioxus somites.

5) The authors perform several highly interesting cross-species comparisons using their
amphioxus scRNA-seq dataset. However, they opted to use broad, pan-embryonic datasets for
zebrafish and clawed frogs that grossly under-represent the mesodermal lineages the authors
are interested in. The authors should re-visit these cross-comparisons with existing datasets that
are more focused. Such comparisons likely also yield more validated, accurate cluster and gene
ontology as the mass-catalog datasets used relied on broad cluster definitions for such niche
lineages like cardiopharyngeal or lateral mesoderm. Comparing with these datasets would
greatly augment and focus the authors' work:

- Nomaru et al., 2021 (Nat. Comms) published a mouse-based single-cell dataset of Tbx1-
expressing cardiopharyngeal lineages.

- Prummel et al., 2022 (Nat. Comms) published a zebrafish lateral plate dataset at the end of
gastrulation.

- De Bono et al., 2023 (again Nat. Comms) build upon Nomaru et al. and provide further details
in their mouse dataset.

We appreciate the reviewer's suggestion to analyse these focused datasets. In a similar vein,
and following other reviewers' comments, we have also added a broader comparison with a
single-cell transcriptomic time course of mouse development (Pijuan-Sala 2019) and another
amphioxus species (*B. floridae*, Ma et al., 2022).

From our global cross-species comparisons, we have learned that the pattern of shared
Transcription Factors (TFs) is a simple and informative measure to identify cell types across
species when dealing with relatively undifferentiated cells (**Fig. 2c and Supplementary Fig.**
**2c**). Thus, we aimed to apply this strategy to the mesoderm-focused datasets suggested by the
reviewer. This allowed us to identify a significant over-representation of shared TFs between
first somite cells of amphioxus and the head mesoderm of zebrafish, which is well aligned with

our results — although it should be noted that this similarity is driven by relatively few TFs, of
 which only TCF21 is actually specifically linked to our proposed scenario (see Figure R2).

First somite pair, left

 **Figure R2:** Jaccard similarity index between the first somites cell cluster in amphioxus and
 various cell populations in the zebrafish dataset, based on shared overexpressed TFs ($FC > 1.5$).

Unfortunately, comparisons with mouse did not yield similarly illuminating results, with no
 pair of cell types having statistically significant TF overlaps in our analyses. In general, we
 have found that global gene expression similarity between our mesodermal populations and the
 cell types identified by Pijuan-Sala (**Supplementary Fig. 3**) is generally lower than with other
 vertebrates, which probably affects the identification of similarities for the focused datasets,
 too.

We would be open to include the focused comparisons between zebrafish and the amphioxus
 mesoderm cell populations. As it stands, however, these similarities are relatively weak and
 merely driven by a handful of TFs that could be best compared in a piecemeal manner. Thus,
 for the sake of clarity and brevity, we think that it is best not to include this analysis and instead
 focus on comparisons against whole-embryo, non-targeted cell atlases, where it is easier to
 appreciate similarities in contrast to dissimilarities.

*** Minor and text points:

220 A) The definitions of lateral plate mesoderm, pharyngeal mesoderm, "head" mesoderm,
 cardiopharyngeal mesoderm, and the likes have become blurred and at times improperly used
 in the literature. The authors' work provides a potentially powerful view point to remedy this,
 and the authors are encouraged to discuss how their data supports (or refutes) differences
 between these different mesoderm "definitions".

We would like to thank the reviewer for this comment but we believe that this discussion goes
far beyond the scope of the study and would merit the writing of a full review so as not to be
superficial and provide real clarification. Indeed, even between different vertebrate species, the
vocabulary, but also the developmental processes, concerning the anterior mesodermal
compartments and the non-axial trunk mesoderm are different. The blurred view we might have
reading the literature is sometimes due to inaccuracies, to the fact that many data obtained in
amniotes have been considered similar in non-amniote species, to the lack of clear data
concerning the homogeneity or heterogeneity of certain mesodermal cell populations, or to a
real problem of definition. We believe that this question still requires more data in various
vertebrate models in order to get a clearer picture.

B) Line 62: Pharyngeal mesoderm also encompasses lateral plate, not solely paraxial.

Thank you for this remark. For clarity, we have changed “paraxial” to “non-axial”.

C) Line 158: ATAC-seq data feels a little out of place here as it is breaking up multiple scRNA-
seq sections.

We understand the concern of the reviewer. However, we choose to place the ATAC-seq
analysis after the global cell atlas as it brings elements to this part of the work and not to the
more focused sub-clusterings. We continue to believe that this is the best way to present data
in an intelligible way.

D) Line 204: Why did the authors chose to describe neural clusters in such detail, seems
misaligned from the general goal of the paper?

We would like to thank the reviewer for this comment. We initially wanted to take the
opportunity to get into the details of the cell atlas even for the neural clusters. However, as it is
not the scope of the work, we have moved this part to supplementary material section. This also
offered us the possibility to have more space to discuss in detail our work and to put it into
perspective with data already published in the literature. We hope that this also partially
alleviates their concerns regarding the structure of the paper (previous point).

E) Throughout the text: please clarify whether “newly described” means by your group in this
study, otherwise properly cite original literature.

We are sorry this was not clear. Each time we wrote “newly described” or “described here” we
meant that the expression pattern was to our knowledge first described in this study. Otherwise,
we have always cited the original literature. However, we realized that we were mistaken
concerning two genes and we modified the manuscript accordingly.

F) Line 301-302, missing word: “The amphioxus orthologues of the latest were also shown to
be expressed in the first somite pair”.

We changed for “It has been shown that the amphioxus orthologues of these are also expressed
in the first pair of somites”.

**Reviewer #2 (Remarks to the Author):**

Grau-Bové et al. constructed an amphioxus neurula stage cell atlas. In general, I have several
comments for improvement.

1. The authors are encouraged to develop an open access website, or upload their data sets to
UCSC cell browser to allow users freely explore the gene expression patterns in different
clusters.

We would like to thank the reviewer for this suggestion. We have created a dedicated web app
to that end: <https://sebelab.crg.eu/amphioxus-neurula-21hpf/>

This tool allows the user to explore the data in multiple ways, e.g. by querying the expression
of specific genes (individually or by gene lists), or identifying lists of markers specific to
precomputed or user-supplied clusters of cells. These functionalities are available for the whole
neurula dataset as well as the sub-clustering analyses of the endoderm, somites and neural cells.

2. In this study, authors employed MARS-seq for scRNAseq library construction. Have the
authors tested other methods, for example 10x chromium or Dropseq, what benefits could
MARS-seq particularly provide? Would the discovery/ statement still hold true if more cells
were sampled?

We now cross-validated our results with data from a different amphioxus species (*B. floridae*,
280 Ma et al., 2022, <https://doi.org/10.1016/j.celrep.2022.110979>), by searching for granular cell
clusters in neurula stage single-cell transcriptomes that match our cell type annotations (see
**Fig. 2**). It is important to consider that this *B. floridae* dataset samples more cells than our
MARS-seq, thus reinforcing the validity of our results.

3. The authors should state the limitation of this study in the discussion section.

We believe the reviewer refers to “Limitation of the study” section that is asked in some journals
at the end of the discussion. This is not the case to our knowledge in *Nature Communications*.

4. In addition to *Ciona intestinalis*, *Danio rerio*, and *Xenopus tropicalis*, authors are suggested
to compare amphioxus atlas with that of human or mouse to reveal the conservation and
divergence of genetic regulatory networks.

We would like to thank the reviewer for the suggestion. We have now added a cross-species
comparison using a single-cell transcriptomic developmental time course for mouse (Pijuan-
Sala et al., 2019, <https://doi.org/10.1038/s41586-019-0933-9>) and for a different amphioxus
species, *B. floridae* (Ma et al., 2022), which also addresses some comments of other reviewers
(see **Fig. 2** and **Supplementary Fig. 3**). Broadly speaking, these new cross-species comparisons
reinforce our results, e.g. the general conservation of TF combinations that define cell types
across chordates (**Fig. 2c**). However, whole-transcriptome comparisons between the *B.*
*lanceolatum* neurula and various mouse cell types and stages revealed a generally poorer level
of conservation, possibly related to the fact that we are comparing relatively undifferentiated
cell populations.

5. The bubble plot in Figure 2a is too crowded.

We have simplified this plot into a heatmap, where it is easier to follow rows and columns, and
significantly enlarged it.

6. More details are required for cross-species homologs identification.

We have tried to give more details in the Methods section concerning this point. Whole-
transcriptome comparisons using SAMap rely on loose gene-gene homology relationships
(defined using BLASTP) that are then refined iteratively. This is now mentioned in line 625.
We have also clarified our procedure to annotate orthologs across species (basically, Broccoli-
based mass orthology assignment + dedicated phylogenies for selected markers and
transcription factors, please see lines 634-639). Orthology assignments and annotations are now
provided as **Supplementary Table 3d**.

7. Previous published amphioxus single cell atlas should be discussed.

We have now included a comparison with the data obtained by Ma and colleagues in the
manuscript as specify above (point 2).

**Reviewer #3 (Remarks to the Author):**

The manuscript by Grau-Bové, is a single cell atlas of the neurula stage in the amphioxus
Branchiostoma lanceolatum. The work exploits publicly available single-cell datasets for
comparisons with other species, and bulk-ATAC seq at three different developmental stages
(two of which were previously published by the same group) to interrogate the regulatory logic
behind the observed cell-specific transcriptional profiles. The authors claim support for the
existence of a transient prechordal plate structure in amphioxus, and anterior mesodermal
regions homologous to the vertebrate cranial/pharyngeal and lateral plate mesoderm. The
manuscript finishes with the proposal of a stepwise scenario for the segregation and remodelling
of the anterior axial mesoderm into prechordal plate, cranial/pharyngeal mesoderm, and lateral
plate mesoderm during vertebrate evolution. Co-option of Pitx2 and Tbx1/10 to the myogenic
program in vertebrates would have been key in this evolutionary transition.

One of the main concerns is the lack of novelty. I cannot find anything in the manuscript that
we did not know before. The Escriva lab already proposed that amphioxus might have
something similar to a prechordal plate in a nice work that combines photoconvertible reporters
and functional assays here: [10.1242/dev.200252](https://doi.org/10.1242/dev.200252). The same lab has also published research on
the evolution of the vertebrate head mesoderm and proposed the same evolutionary scenario
depicted here (see for example [10.1038/s41559-019-0933-z](https://doi.org/10.1038/s41559-019-0933-z)).

We would like to thank the reviewer for acknowledging our previous works. We believe that
bringing new arguments that could support or contradict a hypothetical scenario is critical in
science, even more in evolutionary biology. In our previous works, we proposed a scenario for
the evolution of the vertebrate head mesoderm based on the experiments described by the
reviewer. We aimed at testing this scenario at the level of cell populations by constructing a
cell atlas based on scRNA-seq. This atlas comforts the idea that amphioxus somites at the
neurula stage, before they start to show any morphological differences between
ventral/lateral/dorsal regions, possess several populations of cells with different transcriptomic
profiles. Among them, we could recognize a population that present a profile similar to that of
the lateral plate mesoderm of vertebrates, and a specific population that corresponds to the
anteriormost somites, which shows similarity in terms of gene expression to both pharyngeal
and lateral mesoderm compartments of vertebrates. We also confirm the presence of a

population with endodermal and mesodermal profile in the anterior region that corresponds to
a prechordal plate-like territory. We hence bring new arguments, method and data to support
our hypothetical evolutionary scenario. Moreover, our new analysis of enhancer activity in
zebrafish further clarifies how this evolutionary scenario might have unfolded in a step-by-step
manner (i.e. through the recruitment of new master genes for myogenesis, *Pitx2* and *Tbx1*).

Apart from that, this is not the first single-cell genomics dataset published in amphioxus, and I
can only define as striking, the lack of acknowledgement to the work of Ma and collaborators,
published a year ago (see <https://doi.org/10.1016/j.celrep.2022.110979>). Ma and collaborators
report a total of 148875 cells through 9 developmental stages of the amphioxus *Branchioma*
*floridae*, with the correspondent ATAC-seq, that in this case is also single-cell and not in bulk,
as presented by Grau-Bové and collaborators. Grau-Bové use the same integration method as
357 Ma, so I find hard to believe they have not read the work. This is not only a matter of
358 acknowledgment but a missed opportunity to understand how single-nucleus compares to
359 single-cell profiling, since Ma and collaborators cover the same developmental stage, N3, with
360 a total of 22253 cells, which is a higher number than the number of cells reported by Grau-Bové
at the same stage. Also, a missed opportunity for the authors here, to find support for their own
results. Given that this is not the first single-cell dataset produced in amphioxus, at the very
least, Grau-Bové and collaborators should have studied how their dataset fits within that
previously produced by Ma et al., which is a much more comprehensive study. As it stands, I
cannot see what the Grau-Bové dataset adds to what it was already described by Ma and
collaborators.

We understand the concern of the reviewer. We would like to clarify why we did not compare
our transcriptome with that of Ma et al., in our first submission. Our paper reports the first *B.*
*lanceolatum* neurula stage dataset and Ma et al., also cover this stage in a different species, *B.*
*floridae*. Thus, we agree that cross-species comparisons would have been of great interest as an
independent validation of our results in a different species. Unfortunately, we were not able to
do that right away because not all the necessary information was reported in the original study,
and it was still not available to us at the time of writing our manuscript. Piecing it together
proved to be an unexpected challenge, even with assistance from the authors.

Specifically, the statistical framework we used for cross-species cell type comparisons with
vertebrates (SAMap) required three types of data: (i) cell-by-gene expression matrices for each
stage (these were available); (ii) peptide/transcript sequences from each gene model in the
matrix (unavailable: the gene models used by Ma et al., were not public at the time of their
publication, and were only made available to us a few months ago after we became aware that
they were to be published in a different paper: PNAS, 2023: [10.1073/pnas.2201504120](https://doi.org/10.1073/pnas.2201504120)); and
(iii) reusable cell-level annotations for each cell and stage (again, unavailable in the published
materials and databases: (<https://lifeomics.shinyapps.io/shinyappmulti/>)).

Of course, these limitations would not preclude checking the expression of individual genes
manually in the database released by Ma et al.,. However, in that regard, we were severely
limited by the fact that we could only search genes by symbol (e.g. search for “Hox3”), and Ma
et al., did not provide sufficient details nor the necessary source data for us to be able to work
out which *B. floridae* sequences mapped to these symbols, and, subsequently, obtaining their
orthologs in other species.

Having now overcome these challenges, we have been able to update our study that now include
comparisons with the *B. floridae* transcriptomes. The new analyses provided in this revised
version of the manuscript are as follows:

- • Global alignment of the *B. lanceolatum* neurula transcriptome with various *B. floridae*
stages (from blastula to larva), using SAMap (**Fig. 2**).
- • Transcription Factors (TFs) shared between cell types of multiple chordates, using *B.*
*lanceolatum* as a reference (**Fig. 2**). Again, this shows a remarkable agreement between both
amphioxus species.
- • Cross-checking the existence of specific neurula cell populations in both species, based
on patterns of shared over-expressed TFs. Most cell types described in our dataset can be
matched to specific cell populations in the three neurula datasets covered by Ma et al (N0, N1,
N3; **Supplementary Fig. 2c**).
- • Furthermore, Ma et al., provided a list of marker genes that we can leverage to validate
our cell type annotations. Many of them were already reported in the original version of the
manuscript, and the ones that were missing, as pointed out below by the reviewer, have now
been included in **Supplementary Fig. 2b**.

Finally, while we agree that assessing the methodological strengths and limitations of snRNA-
seq and scRNA-seq technologies is a most interesting question by itself, we do not think that
this is the best setup for such a benchmarking study — the main reason being the fact that we
would be analysing two different species (separated by ~100 Mya of independent evolution),
which is a major confounding factor.

An even major concern is the number of imprecisions throughout the manuscript. The use of
metacells is clever, especially if you have sparse data, but it also indicates the dataset lacks
granularity. This is not a negative per se, but a limitation that the authors should acknowledge
by: i) avoiding fine resolution annotations;

In order to accommodate multiple levels of granularity/resolution in our analyses, we annotate
cells at multiple levels: metacells (high-granularity clustering of statistically identical cells,
resulting in 176 clusters), cell types (i.e. grouping metacells into cell types using shared
markers), and subset reclusterings of cells corresponding to specific cell types (e.g. new
metacells to find additional granularity in the broadly defined somites).

Expression patterns at these three levels of annotation are reflected in **Fig. 1, 4 and 5**, for
example. For added flexibility, they can now be browsed in our interactive database as well:
<https://sebelab.crg.eu/amphioxus-neurula-21hpf/>

Among other things, this tool allows the user to identify the best markers for any cell clustering
scheme (either our precomputed cell types or any user-supplied cluster), in a very flexible
manner.

ii) or by validating their proposed annotations using orthogonal approaches. This could well be
in situ, but not the sort of in situ included in this study, which do not offer single-cell
resolution nor co-expression information, more like the sort of in situ the Escriva lab published

previously here: doi:10.1242/dev.200252. Because the authors have not done one of the other
(i or ii), the annotations and conclusions are extremely confusing.

We apologize but we also do not understand what the reviewer means by “sort of”. Does the
reviewer mean chromogenic *in situ* hybridization? Fluorescent *in situ* hybridization?

We have added new multiple *in situ* hybridization using the HCR method to bring an
“orthogonal approach” validating the annotation of the prechordal plate-like territory, the
anteriormost somite territory and the ventral part of the somite territory. These data are now
included in **Fig. 4 and 5**.

Furthermore, as suggested earlier by the reviewer, we have also validated our cell type
annotations by means of cross-species comparisons with the *B. floridae* dataset
(**Supplementary Fig. 2c**). Specifically, we reclustered each neurula stage of the *B. floridae*
dataset into metacells (N0, N1, N3; we used the same procedure as for the *B. lanceolatum*
dataset), and evaluated the transcriptional similarity between our *B. lanceolatum* cell types and
the best-matching metacells in *B. floridae*, measuring shared TF expression across species.
Essentially, this analysis determines whether homologous cell populations characterized by
specific TF combinations can be identified in the two species (top markers listed in
**Supplementary Fig. 2c**). We thus validate the major cell types of the endoderm, mesoderm
(except the putative neuro/endo metacell, which lacks specific TFs in our dataset), and ectoderm
layers (except one of the epidermal/neural metacells).

A few examples of such imprecisions are detail below:

1. Mesoderm annotations do not coincide with the profiles previously published by the Escriva
or Yu labs. Works from the two labs showed that mesoderm is highly regionalised anterior-
posteriorly, dorso-ventrally and medio-laterally, with specific populations expressing particular
combinations of genes. Where are all these cells here? Do the authors confirm? Deny? Or none
of the above? The work does not provide further understanding on this matter. Furthermore, the
work of the Yu lab is absent from the reference list (10.3389/fcell.2021.607057).

We are sorry we do not understand reviewer’s comment. We indeed showed that we could
define 12 cell populations in the somitic compartment (see **Fig. 5** and result section), 10 of
which we could assign to anterior somites, posterior somites, middle somites, differentiating
muscular part of the somites, left and right populations, tailbud populations, etc. We have added
the reference suggested by the reviewer and discussed our results in the light of the proposition
of somite compartmentalization made by Yong and colleagues (10.3389/fcell.2021.607057),
please see lines 303.

2. How can netrin be localised to metacell 9 only, in the posterior neural tube, when it is
expressed along the entire neural tube? See Shimeld, 2000. Netrin must be also in 14, 8, 20 and
21, probably in 12 too. This is surely a phrasing problem. It might be helpful to have the top
differentially expressed genes per metacell, to see the different combinations. Overexpression
is not always differential expression. As it is written it sounds like a guessing based on the
expression of individual genes. The authors submitted a table with the re-clustering, but it does
not detail how the profiles have been refined.

We again are not entirely sure we understand the reviewer's comment. Our aim was not to
describe the expression of genes but to highlight the genes the expression of which allows us
to assign a cell cluster to an embryonic territory. Indeed *Netrin* is not expressed in metacell 9
only, as even stated in the sentence that follows "According to the expression of the floor plate
marker genes *Chordin*, *Foxaa*, *Goosecoid*, *Netrin*, *Nkx2.1* and *Nkx6*^{1,9-15} (**Supplementary Fig.**
**5**), metacells 8 and 14 could be assigned to this structure, with metacell 8 additionally
expressing the posterior genes *Cdx* and *Hox3*^{1,6,16} (**Supplementary Fig. 5**).". The description
of the data on the neural subclustering are now presented as **Supplementary Materials**.

We have always detailed for each metacell the combination of genes that enabled us to assign
clusters to a territory, this is not a guess based on the expression of a single gene and hope that
it is clearer in this new version of the manuscript. We now moreover provide an interactive
version of our dataset through our web app, which can be used to check patterns of coexpression
in a flexible manner: <https://sebelab.crg.eu/amphioxus-neurula-21hpf/>

3. Another case is that of the epidermis, which of the three classifications are *FoxJ* or *Keratin*
or none of the above? This does not allow relating this annotation to the clusters defined by Ma
and collaborators, and the works published previously on epidermis gene expression. It is very
confusing.

Following the reviewer's suggestion, we have added the expression profile of *Foxj1* and various
*Keratin* genes, as well as other markers used by Ma et al., to the new **Supplementary Figure**
**2b**. These genes are widely expressed in the two main epidermal clusters in our dataset
(anterior/posterior), as well as in the epidermal/neural cells. This result recapitulates the results
by Ma et al, who found that these two markers were broadly expressed in the single
epidermis/ectoderm cell cluster they defined in their study (unfortunately, finer-grained
annotations of the epidermis were not available).

Beyond these markers, it is worth noting that our annotation of the epidermal clusters is
supported by the overexpression of other key genes: *AP2*, *Dlx*, *Grhl*, and *Fhl2* (**Supplementary**
**Figure 2b** and **Supplementary Figure 4a**).

4. I do not understand either, the proposal that metacells 4 and 22 correspond to the
Hypothalamo-prethalamic primordium. Since Albuixech-Crespo and collaborators decided to
rename the amphioxus forebrain as Hypothalamo-prethalamic primordium at this stage of
development (7ss, N3), there has been several publications that demonstrate this does not reflect
how the amphioxus brain is structured or develops. The amphioxus brain at this stage has not
even started the process of neurogenesis. It has been shown that embryos at the 7ss (N3) do not
have hypothalamus nor hypothalamus primordium, that the hypothalamus forms by intercalated
growth and addition of new cells at the 7ss and these progenitor cells do not have a recognisable
hypothalamic profile until the 11-12ss, when they start expressing *Otp* (see
<https://doi.org/10.3389/fnins.2022.812223>). Most of the hypothalamic centres in vertebrates
depend on *Otp* expression (see doi:10.1242/dev.139055), so the annotations here does not make
any sense. *Fezf* is involved in the formation of serotonergic cells of the frontal eye complex in
amphioxus, that would correspond to the pre-optic/optic area, which is possibly what the

authors see here. *Six3/6* and *Otx*, on the other side, are expressed in very distinct parts of the
amphioxus brain, also in vertebrates. There are numerous publications backing this up. These
cannot be used on their own to have such refine annotation. *Ctfp1*, has previously been reported
at a single-cell resolution in situ (see <https://doi.org/10.1101/2023.03.13.532359>) and it is
expressed in two different brain regions, excluding the hypothalamus. This is the reason why
Grau-Bové see cells expressing this marker at this stage, because it is not expressed by the
hypothalamus. Note that the gene is described as being expressed in the cerebral vesicle as a
whole, which is incorrect. The stage analysed by Grau-Bové is too young and the sequencing
is too shallow to ascribe these metacells to such particular brain areas. The cross-species
comparisons in Figure 2, also show this classification to be arbitrary, as what they call
Hypothalamo-prethalamic primordium is hitting everywhere in the anterior nervous system of
zebrafish and xenopus. Perhaps annotating these cells as anterior and posterior forebrain would
be more appropriate in this case.

It may seem that the reviewer misinterpreted some aspects of the publication by Albuixech-
Crespo et al 2017. Even though this publication is not being reviewed here, and it has little to
do with the current study, we would like to provide some clarifications.

First, Albuixech-Crespo et al. proposed a model to explain the regionalization of the developing
amphioxus brain. The proposed model is based on the general prosomeric model, which implies
several ontogenetic rules that have to be met (see page 2 in Albuixech-Crespo et al. 2017). One
important concept here is that, earlier in development, broad areas (or primordia) are specified
and then they are subsequently enlarged (even if by intercalary growth) and split into
subregions. Other studies further elaborate on this model and provide additional information on
its basic concepts (Puelles et al., 2013; Niewenhuys and Puelles, 2013; Puelles 2013; Puelles
and Rubenstein 2015).

Second, Albuixech-Crespo et al. 2017 has not renamed the forebrain as "Hypothalamo-
prethalamic primordium"; if anything, it proposes that "forebrain", in vertebrates, should also
include the midbrain (and, in fact, that archencephalon is a more accurate term than forebrain
for that region at that stage). Then, it reports two clear molecular partitions within the
archencephalon in amphioxus at the studied stage, shared with vertebrates, and provides names
for the identities of such partitions based on the topological comparison with the developing
vertebrate neural tube.

Third, by no means the proposed model in Albuixech-Crespo et al. depends on whether or not
there is neurogenesis at the studied stage or whether there is a functional (!?) hypothalamus or
even cells already committed to become hypothalamic neurons. This is not what the model is
about and it is a misconception. What the model allows is to define some regions at a given
stage, from which certain structures will eventually develop. Thus, we do not see any issue in
using this well validated model of regionalization to topologically map our metacells.

Fourth, the argument provided about the expression of specific genes again do not fit with a
proper understanding of the model proposed by Albuixech-Crespo et al (or comparative
neuroanatomy in general). For instance, the fact that most of the hypothalamic centers in
vertebrates depend on *Otp* expression (later in development) has nothing to do with the validity
of the defined regions at the studied stage (much earlier in development). Even in vertebrates,
the hypothalamic region can be well recognized before *OTP* expression is observed in
postmitotic cells (Puelles et al., 2012; Puelles and Rubenstein 2015; Ferran et al 2015).

Development is a dynamic process and gene expression markers should not be used as proofs
(or disproofs) of homology without the adequate time and topological references (something
far too common in the traditional amphioxus literature, unfortunately). This is particularly the
case when mixing terminally differentiated regions or structures with regions from early
developing neural plates.

Irrespective of these considerations with respect to the model proposed by Albuixech-Crespo
et al and used here to topologically map our metacells, we have now moved this comparison to
Supplementary Information and Supplementary Fig. 5. to focus more on the main messages of
the paper.

5. In the ATAC-seq section Grau-Bové also proposes elements of hypothalamic-specific
expression of *Soxb1c*(*Sox2*) and *Pou3fl* (*Pou3f4*), which we know for certain they are not
specific of the hypothalamus but more general central nervous system markers, also at the N3
stage, see doi: 10.7150/ijbs.3.356, and <https://doi.org/10.3390/cells12040614>.

We would like to thank the reviewer for noticing this imprecision. We actually meant that, in
the hypothalamus, *Soxb1c* shows a combined enrichment of expression and motif usage
(likewise for *Pou3fl* in the neural tailbud and Di-Mesencephalic cells). We have amended the
text to clarify this, please see lines 218-222.

6. I cannot find clarity either about the prechordal plate (metacell 1) is this co-expressing
*Gooseoid*, *Dmbx* and *Nodal*, as proposed earlier? Is this *Bra* positive or not? Because the
vertebrate prechordal plate is not? And does this express any FGF receptors that could explain
the phenotype described by the Escriva lab before (10.1242/dev.200252)?

As explained in the manuscript (from line 270), the metacell we assigned to the prechordal
plate-like structure expresses *Dmbx* and *Brachyury2* among others. According to the article
mentioned by the reviewer ((10.1242/dev.200252)), at the stage studied here, *Gooseoid* is no
longer expressed in this region, and *Nodal* expression is getting restricted to the left
somites/tailbud. As also discussed in our previous work, the vertebrate prechordal plate indeed
does not express *Brachyury*, but *Gooseoid*, which is a repressor of *Brachyury*. We proposed
in the corresponding article that the duplication of *Brachyury* in amphioxus might have allowed
one of the duplicate (*Brachyury2*) to keep on being expressed in the prechordal plate-like
territory while *Gooseoid* expression fades during neurulation, giving this peculiar ability to
this region to grow by intercalation, resulting in this derived character that is the “notochord in
the head” of cephalochordates.

Concerning FGFR, there is only one gene in amphioxus which we previously showed is
expressed at the time at which its inhibition by SU5402 affects the formation of the anterior
mesoderm (during gastrulation) in the mesendoderm (doi: 10.1073/pnas.1014235108). The
phenotype observed in (10.1242/dev.200252) also corresponds to an inhibition of the FGF
receptor during gastrulation and not during neurulation.

7. The reporter expression in zebrafish should be accompanied by co-detection of GFP with the
genes they claim characterise those specific tissues where the reporter is expressed. Otherwise,
this only shows these enhancers that are able to drive expression in zebrafish.

We understand the concern of the reviewer. To improve the analysis of GFP expression in
transgenic zebrafish embryos, and as also suggested by reviewer 1, we undertook a double HCR

experiment with probes against *GFP* and *Prrx1a*, which has been described as expressed in the
head mesoderm and finbuds. The results show an overlap of labelling in both the finbuds and
part of the head mesoderm (see **Supplementary Fig.8**).

There are also some other technical details that are puzzling, 5% of cell recovery and cell types
captured in the same proportions as they are in the embryos? It is of course expected to have a
low representation of minority cell types, but not following the same ratios. Capture should be
random after dissociation. It looks like embryos were partially digested, which would explain
why there are so many epidermal cells: they are the first ones to peel off. It is hard to say because
there are no quality controls in the paper.

We are not sure we understand what the reviewer means by 5% of cell recovery. We suspect
there has been a misunderstanding, as we have sampled 14,586 single-cell transcriptomes for a
stage at which the embryo contains ~3,000 cells, and therefore each cell state should be sampled
at ~5x coverage, rather than 5% (line 126).

As a quality control on our cell sampling strategy, we showed that among these transcriptomes,
the proportion assigned to each tissue/layer corresponds to the proportion observed in the
embryo (**Fig. 1d**), highlighting the quality of the data and the little bias in cell type recovery
after dissociation. The proportion of transcriptomes assigned to the epidermal territory is indeed
high but similar to the proportion of epidermal cells found at this stage in the amphioxus
embryo. To clarify this point, we now mention the number of sampled cells in each experiment
in **Fig. 1d**. We also provide further quality controls for our sampling and clustering of single
cell transcriptomes in **Supplementary Fig.1a**.

REVIEWER COMMENTS

Reviewer #1 (Remarks to the Author):

In the revised manuscript by Grau-Bové, Subirana, et al., the authors refined their manuscript considerably based on the input of the peer reviewers. The overall context and message of the manuscript is much better embedded in the field's current knowledge (albeit the concern of novelty and overlap with previous work, also raised by other reviewers).

Several of the implemented changes are laudable, such as Fig. 6 with Alx staining is instrumental. Their outline concerning the nomenclature definitions in the literature (or lack thereof) is a critical take-home message that would deserve emphasis in the manuscript. The abstract would still benefit from stronger, more concise wording to emphasize the key take-home message(s) of the manuscript.

Nonetheless, in particular the zebrafish-based reporter assays remain a critical issue of the current manuscript.

The authors added now HCR on fixed reporter-GFP embryos for two genes and argue about specificity. The reviewer definitely acknowledges the challenges of after-the-fact experiments. Yet, the limited additional data and written-up reasoning do not eliminate the basic concern about the used transgenesis marker and pleiotropic expression.

Specifically, the HCR/GFP combinations for the Gata1/2/3 and Tbx1/10 elements is difficult to decipher and look rather pleiotropic. While there is certainly overlap in the areas of endogenous expression, lots of other GFP signal is present that renders claims of specificity challenging.

Together, these zebrafish reporter experiments are executed with sub-par tools and documentation standards that this reviewer does not regard as the state-of-the-art in the field and as necessary to substantiate the made claims.

Concerning comparisons to other species, the authors seem to focus solely on transcription factor genes (R2 for instance). Expanding this as a wider net for comparative studies would likely expand their grasp of the evolutionary depth of their investigated biology, as well as reveal likely instrumental markers for future studies as well.

Reviewer #2 (Remarks to the Author):

The website is not accessible at both Google Chrome and Firefox browser, please fix the problem

<https://sebelab.crg.eu/amphioxus-neurula-21hpf/>

Proxy Error

The proxy server could not handle the request GET /amphioxus-neurula-21hpf/.

Reason: Error during SSL Handshake with remote server

Reviewer #3 (Remarks to the Author):

Overall, I believe that the revised version of the manuscript is significantly improved. It provides a clearer context for the previous research conducted and effectively shows how these findings align with the presented results. The scRNA-seq browser is user-friendly and easy to navigate, allowing for a comprehensive understanding of the data. Furthermore, the inclusion of *B.floridae* data for comparison purposes is very valuable.

The addition of HCRs is also commendable as it enhances the usefulness of the findings. Given this, I would suggest that the authors consider removing the "a" panel in figures 4, 5 and supplementary figures 5, 6, and 7. These panels can be misleading since scRNA-seq is not a spatial technique. Therefore, the specific localization of the metacells over the drawing of a neurula is purely interpretive and could lead to misinterpretations about the experimental design employed. Instead, using the force-directed graph utilized in Figure 1b and highlighting the relevant metacells would be a more appropriate reference for readers.

My concern regarding the hypo/prethalamus annotation still remains. It is important to note that the hypothalamus does not exist at this specific developmental stage. One cannot sequence something that does not exist. The authors argue that the progenitors are present and later differentiate, but division arrest + chase experiments in amphioxus indicate otherwise. Hypothalamic precursors are not generated until later stages and only mature during the early larval stages. Consequently, this raises doubts about all obtained results, either: 1) the experiment was poorly conducted (which I do not believe is the case, given the reputable participating labs), or 2) the interpretation of the results is inaccurate. Supporting the latter explanation are two observations: 1) Figure 2C does not demonstrate any correlation with the hypothalamus in vertebrates (or even in ciona, were it is particularly challenging to interpret), and 2) Supplementary Figure 2c reveals that the transcription factors associated with the annotated hypo/prethalamus in both *B.lanceolatum* and *B.floridiae* do not appear to be directly involved in hypothalamic development. For instance, RAX is known as a retinal marker, INSM2 has no established link to hypothalamus development, and the remaining factors are too broadly expressed to be specific for the hypothalamus. To appeal to a wider audience, it may be more interpretable to label these metacells as part of the forebrain, a term that is universally understood. This would strengthen the comparisons to vertebrates shown in Figure 2C and support the general idea that scRNA-seq in amphioxus has sufficient power to enable meaningful species comparisons.

Regarding Supplementary Figure 8, the in-situ hybridization results are visually appealing, but it is unclear how to interpret the mostly mutually exclusive expression patterns of GFP and PRRX1a.

In sum, I appreciate the value of this study. However, the study does not introduce new mechanisms, molecules, concepts, or techniques. The scRNA-Seq protocol used here was previously published by Sebe-Pedros (doi: 10.1016/j.cell.2018.05.019). Many previous studies from laboratories including the Holland and Garcia-Fernandez labs (Yu lab also discussed previously) have already demonstrated that somites exhibit molecular regionalization before they become morphologically distinct. There is previous published evidence, brought up by the Escriva lab itself (as previously discussed), suggesting the presence of structures analogous to a prechordal plate and lateral plate mesoderm in amphioxus. Furthermore, it has been consistently shown in most published papers at this stage of development that the transcriptional profile of the anterior endomesoderm appears mixed. How all this is controlled, is the key question that remains unanswered.

**Please find below our point-by-point answer to the reviewer's comments. For clarity we've**
**taken the initiative of numbering the comments and our answers are in blue.**

**Reviewer #1 (Remarks to the Author):**

In the revised manuscript by Grau-Bové, Subirana, et al., the authors refined their manuscript
considerably based on the input of the peer reviewers. The overall context and message of the
manuscript is much better embedded in the field's current knowledge (albeit the concern of
novelty and overlap with previous work, also raised by other reviewers).

- 1. Several of the implemented changes are laudable, such as Fig. 6 with Alx staining is
instrumental. Their outline concerning the nomenclature definitions in the literature (or
lack thereof) is a critical take-home message that would deserve emphasis in the
manuscript. The abstract would still benefit from stronger, more concise wording to
emphasize the key take-home message(s) of the manuscript.

We thank the reviewer for these positive comments. To be more concise in the abstract, we
have replaced the sentence L34 « *Our data allowed us to confirm the presence of a prechordal-*
*plate like territory in amphioxus, and shows that cell populations of the anteriormost somites*
*and of the ventral part of the somites present a transcriptomic profile supporting the homology*
*with vertebrate cranial/pharyngeal and lateral plate mesoderm* » by « *Our data allowed us to*
*validate the presence of a prechordal-plate like territory in amphioxus. Additionally, the*
*transcriptomic profile of somite cell populations supports the homology between specific*
*territories of amphioxus somites and vertebrate cranial/pharyngeal and lateral plate*
*mesoderm.* »

- 2. Nonetheless, in particular the zebrafish-based reporter assays remain a critical issue of
the current manuscript.

The authors added now HCR on fixed reporter-GFP embryos for two genes and argue
about specificity. The reviewer definitely acknowledges the challenges of after-the-fact
experiments. Yet, the limited additional data and written-up reasoning do not eliminate
the basic concern about the used transgenesis marker and pleiotropic expression.

Specifically, the HCR/GFP combinations for the Gata1/2/3 and Tbx1/10 elements is
difficult to decipher and look rather pleiotropic. While there is certainly overlap in the
areas of endogenous expression, lots of other GFP signal is present that renders claims
of specificity challenging.

Together, these zebrafish reporter experiments are executed with sub-par tools and
documentation standards that this reviewer does not regard as the state-of-the-art in the
field and as necessary to substantiate the made claims.

We understand the concern of the reviewer. There might have been a misunderstanding
concerning our interpretation of the transgenic results and the objectives of such experiments.
The data we generated were aimed to test if some putative regulatory elements of amphioxus
genes were able to drive in a vertebrate the expression (not exclusively) of a reporter gene in
head or lateral plate mesoderm and derivatives. In other words, we wanted to check whether
enhancers putatively present in the chordate ancestor that evolved a given function in
amphioxus could have been selected during evolution for a function in a tissue that appeared
specifically in vertebrates (the head and lateral mesoderm). We did not aim to undertake a

detailed enhancer functional study. The overlap, which is not complete, of *Prrx1a* expression
and reporter expression provided in the HCR experiment, shows that the regions tested drive
expression in the zebrafish head mesoderm and finbud, although it shows they also drive
expression in other territories in the head. This suggests that the orthologs of the genes tested
here, present in the chordate ancestor, had the capability to be recruited during vertebrate
evolution for expression in head mesoderm and LPM. However, we decided, as suggested by
the editor, to move the transgenic data to supplementary figures and to shorten drastically the
corresponding result section.

3. Concerning comparisons to other species, the authors seem to focus solely on
transcription factor genes (R2 for instance). Expanding this as a wider net for
comparative studies would likely expand their grasp of the evolutionary depth of their
investigated biology, as well as reveal likely instrumental markers for future studies as
well.

We usually employ all cell type-specific orthologous genes for cross-species comparisons. But
comparing developmental stages offers a unique challenge: both undifferentiated and terminal
cell types co-exist in the sampled embryos, and they have extremely different transcriptome
profiles. Terminal cell types contain dozens or hundreds of effector genes specifically
expressed, while undifferentiated cells express a much smaller set of genes, often only
combinations of transcription factors. To account for this asymmetry and to avoid making ad
hoc decisions about the differentiation state of each cell cluster, we focused our comparisons
only on transcription factors. This is the practical reason for this analysis, albeit actually many
authors have suggested one should focus primarily on transcription factors (cell type regulatory
identity), instead of effector genes (cell type function) for cross-species comparisons and, thus,
this is common practice in the field.

**Reviewer #2 (Remarks to the Author):**

1. The website is not accessible at both Google Chrome and Firefox browser, please fix
the problem <https://sebelab.crg.eu/amphioxus-neurula-21hpf/> Proxy Error The proxy
server could not handle the request GET /amphioxus-neurula-21hpf/. Reason: Error
during SSL Handshake with remote server

We apologize for the inconvenience. The server was down for a week in late December for
technical reasons, but it is now accessible again.

**Reviewer #3 (Remarks to the Author):**

Overall, I believe that the revised version of the manuscript is significantly improved. It
provides a clearer context for the previous research conducted and effectively shows how these
findings align with the presented results. The scRNA-seq browser is user-friendly and easy to
navigate, allowing for a comprehensive understanding of the data. Furthermore, the inclusion
of *B.floridae* data for comparison purposes is very valuable.

1. The addition of HCRs is also commendable as it enhances the usefulness of the findings.
Given this, I would suggest that the authors consider removing the "a" panel in figures
4, 5 and supplementary figures 5, 6, and 7. These panels can be misleading since
scRNA-seq is not a spatial technique. Therefore, the specific localization of the
metacells over the drawing of a neurula is purely interpretive and could lead to
misinterpretations about the experimental design employed. Instead, using the force-
directed graph utilized in Figure 1b and highlighting the relevant metacells would be a
more appropriate reference for readers.

We understand the concern of the reviewer. We have modified the panels "a" from the figures
mentioned so that not to be misleading. The aim of these panels was to help the readers who
are not familiar with amphioxus embryo to better understand the description we made
concerning the transcriptomic profiles of cell populations. The panels "a" now only show
schemes that we believe are useful and convenient.

2. My concern regarding the hypo/prethalamus primordium annotation still remains. It is
important to note that the hypothalamus does not exist at this specific developmental
stage. One cannot sequence something that does not exist. The authors argue that the
progenitors are present and later differentiate, but division arrest + chase experiments
in amphioxus indicate otherwise. Hypothalamic precursors are not generated until later
stages and only mature during the early larval stages. Consequently, this raises doubts
about all obtained results, either: 1) the experiment was poorly conducted (which I do
not believe is the case, given the reputable participating labs), or 2) the interpretation of
the results is inaccurate. Supporting the latter explanation are two observations: 1)
Figure 2C does not demonstrate any correlation with the hypothalamus in vertebrates
(or even in ciona, were it is particularly challenging to interpret), and 2) Supplementary
Figure 2c reveals that the transcription factors associated with the annotated
hypo/prethalamus primordium in both *B.lanceolatum* and *B.floridiae* do not appear to be
directly involved in hypothalamic development. For instance, RAX is known as a retinal
marker, INSM2 has no established link to hypothalamus development, and the
remaining factors are too broadly expressed to be specific for the hypothalamus. To
appeal to a wider audience, it may be more interpretable to label these metacells as part
of the forebrain, a term that is universally understood. This would strengthen the
comparisons to vertebrates shown in Figure 2C and support the general idea that
scRNA-seq in amphioxus has sufficient power to enable meaningful species
comparisons.

We thank the reviewer for this remark aimed at helping us presenting the data in a more
appealing way for the broad audience of the journal. However, the term "forebrain" corresponds
to a vertebrate structure, and it could be misleading to name a presumptive amphioxus
embryonic structure by the name of a non-homologous vertebrate structure. Following the
reviewer's comment and the editor's recommendation we have hence decided to modify the
name « Hypothalamo-prethalamus primordium » to « anterior archencephalon » in the figures
(Fig. 1 and Supplementary Figures) and text (one occurrence in Supplementary Material, L7).

3. Regarding Supplementary Figure 8, the in-situ hybridization results are visually
appealing, but it is unclear how to interpret the mostly mutually exclusive expression
patterns of GFP and PRRX1a.

The figure shows that the *GFP* expression partly overlaps with the one of *Prrx1a*, which is
expressed in the head mesoderm and finbuds. This shows that the putative enhancers tested are
able to drive, although not exclusively, the expression of the reporter gene in these regions.

4. In sum, I appreciate the value of this study. However, the study does not introduce new
mechanisms, molecules, concepts, or techniques. The scRNA-Seq protocol used here
was previously published by Sebe-Pedros (doi: 10.1016/j.cell.2018.05.019). Many
previous studies from laboratories including the Holland and Garcia-Fernandez labs (Yu
lab also discussed previously) have already demonstrated that somites exhibit molecular
regionalization before they become morphologically distinct. There is previous
published evidence, brought up by the Escriva lab itself (as previously discussed),
suggesting the presence of structures analogous to a prechordal plate and lateral plate
mesoderm in amphioxus. Furthermore, it has been consistently shown in most published
papers at this stage of development that the transcriptional profile of the anterior
endomesoderm appears mixed. How all this is controlled, is the key question that
remains unanswered.

The reviewer is perfectly right, the protocol was already published. The aim of the work
presented here was to shed light on mesoderm evolution in chordates by using a scRNA-seq
approach. As we already stated, we believe that bringing new arguments that could support or
contradict a hypothetical scenario is critical in science, even more in evolutionary biology. We
would also argue that for the first time, by using the scRNA-seq atlas we generated, we were
able to identify a transcription factor (*Gata1/2/3*) that could be a key player in the development
of the non-myotomal part of the somites in amphioxus, and we demonstrated by *in vivo*
functional analysis that it is indeed the case. We believe that our approach, while reinforcing
previous knowledge, also brings new insights into how mesoderm may have evolved at the
level of cell populations in the chordate lineage.

REVIEWERS' COMMENTS

Reviewer #1 (Remarks to the Author):

In response to the authors' rebuttal points:

1. The provided (limited) still images indeed show enhancer activity in cells with PRRX1 - nonetheless, the interpretation of these data remains challenging, also as the provided images are at single timepoints and interpretation of expression domains bases on relative anatomical positions.

Further, and connecting to the initially raised concerns, the transgenesis marker used in these assays obscures major parts of the head. Given the transgenesis marker has the same fluorescence as the tested reporters, cross-talk between these in cis elements remains a concern.

Deducing that a) activity in the desired tissues is specific and b) activity in other tissues is a consequence of evolutionary divergence is a select way of interpreting these experiments. Without further controls and comparisons with other enhancers to be tested, another interpretation is promiscuity of enhancers when tested in different species. Here again, time course images would have greatly helped with the interpretation, and the inclusion of additional candidates.

Importantly, as the authors stated, the zebrafish lines are no longer available, raising concerns about reproducibility.

2. While the focus on transcription factors as cluster-defining or lineage-determining readouts is understandable and indeed commonly used, it severely limits the interpretation of the authors' beautiful data. Uncovering broader signatures for individual cell types and lineages is a major challenge in the field and a missed opportunity in the current presentation (which the reviewer hopes will be caught up on in future projects).

**Please find below our point-by-point answer to the reviewer's comments. For clarity our**
**answers are in blue.**

**Reviewer #1 (Remarks to the Author):**

1. The provided (limited) still images indeed show enhancer activity in cells with PRRX1 -
nonetheless, the interpretation of these data remains challenging, also as the provided images
are at single timepoints and interpretation of expression domains bases on relative anatomical
positions.

Further, and connecting to the initially raised concerns, the transgenesis marker used in these
assays obscures major parts of the head. Given the transgenesis marker has the same
fluorescence as the tested reporters, cross-talk between these in cis elements remains a concern.

Deducing that a) activity in the desired tissues is specific and b) activity in other tissues is a
consequence of evolutionary divergence is a select way of interpreting these experiments.
Without further controls and comparisons with other enhancers to be tested, another
interpretation is promiscuity of enhancers when tested in different species. Here again, time
course images would have greatly helped with the interpretation, and the inclusion of additional
candidates.

Importantly, as the authors stated, the zebrafish lines are no longer available, raising concerns
about reproducibility.

We understand the concern of the reviewer, and we are aware that the inability to carry out new
experiments due to the fact that we had to sacrifice the transgenic lines is damaging. With this
in mind, we have added the following sentence "However, these data still need to be supported
by a more precise analysis of reporter expression domains at different developmental stages."
line 383 and line 415 of the "Results and Discussion" section. We also added in the Methods
section line 754 "The transgenic lines are no longer available"

To clarify a possible misunderstanding, the experiment using transgenics is not intended to
establish that a given enhancer is "specific" or has "divergent" activity. The use of this type of
approach (used in numerous other studies) is simply to test (not to prove) if an ancestral
enhancer similar to the one present today in amphioxus, could already possess elements that
had the capacity to direct expression in structures that did not exist in the original organism but
that appeared during the course of evolution. This type of heredity is the basis of co-option in
the regulation of gene expression.

2. While the focus on transcription factors as cluster-defining or lineage-determining readouts
is understandable and indeed commonly used, it severely limits the interpretation of the authors'
beautiful data. Uncovering broader signatures for individual cell types and lineages is a major
challenge in the field and a missed opportunity in the current presentation (which the reviewer
hopes will be caught up on in future projects).

We would like to thank the reviewer for this positive comment. We understand the reviewer
frustration at the fact that the data in the cell atlas may not have been exploited in sufficient
depth. We obviously intend to continue exploiting the data, and we hope that the access offered

to the community through the user friendly shiny app will also enable others to browse the data
to answer many questions.
